# The RNA-binding protein HuR modulates the expression of the disease-linked *CCL2* rs1024611G-rs13900T haplotype

Feroz Akhtar[1], Joselin Hernandez Ruiz[2], Ya-Guang Liu[3], Roy G Resendez[1], Denis Feliers[4†], Liza D Morales[5], Alvaro Diaz-Badillo[1], Donna M Lehman[1], Rector Arya[1], Juan Carlos Lopez Alvarenga[6], John Blangero[5], Ravindranath Duggirala[1], Srinivas Mummidi[1]*

[1]Department of Health and Behavioral Sciences, Texas A&M University-San Antonio, San Antonio, United States; [2]Utah Center for Genetic Discovery, Department of Human Genetics, University of Utah, Salt Lake City, United States; [3]Department of Pathology, School of Medicine, University of Texas Health San Antonio, San Antonio, United States; [4]Department of Medicine, School of Medicine, University of Texas Health San Antonio, San Antonio, United States; [5]South Texas Diabetes and Obesity Institute, Department of Genetics, School of Medicine, University of Texas Rio Grande Valley, Brownsville, United States; [6]Department of Population Health and Biostatistics, School of Medicine, University of Texas Rio Grande Valley, Harlingen, United States

*For correspondence:
Srinivas.Mummidi@tamusa.edu

Present address: †Novartis Institute for Biomedical Research San Diego, San Diego, United States

Competing interest: The authors declare that no competing interests exist.

## eLife Assessment

CCL2 is a chemokine with immune cell chemoattractant properties, and it appears to play a role in several chronic inflammatory diseases. The RNA-binding protein HuR controls the stability and translation of CCL2 mRNA. This paper presents **convincing** evidence that a relatively common genetic variant tied to several disease phenotypes affects the interaction between the mRNA of CCL2 and the RNA-binding protein HuR. While the experiments cannot definitively distinguish between effects on RNA transcription and stability, CCL2 is thought to be relevant for leukocyte migration in various conditions, including chronic inflammation and cancer, and the study presents **important** findings that may be relevant to a broad audience.

## Abstract

CC-chemokine ligand 2 (CCL2) is involved in the pathogenesis of several diseases associated with monocyte/macrophage recruitment, such as HIV-associated neurocognitive disorder (HAND), tuberculosis, and atherosclerosis. The rs1024611 (alleles: A>G; G is the risk allele) polymorphism in the *CCL2 cis*-regulatory region is associated with increased CCL2 expression in vitro and ex vivo, leukocyte mobilization in vivo, and deleterious disease outcomes. However, the molecular basis for the rs1024611-associated differential CCL2 expression remains poorly characterized. It is conceivable that genetic variant(s) in linkage disequilibrium (LD) with rs1024611 could mediate such effects. Previously, we used rs13900 (alleles: C>T) in the *CCL2* 3'untranslated region (3' UTR) that is in perfect LD with rs1024611 to demonstrate allelic expression imbalance (AEI) of *CCL2* in heterozygous individuals. Here, we tested the hypothesis that the rs13900 could modulate *CCL2* expression by altering mRNA turnover and/or translatability. The rs13900 T allele conferred greater stability to the *CCL2* transcript when compared to the rs13900 C allele. The rs13900 T allele also had increased binding to Human Antigen R (HuR), an RNA-binding protein, in vitro and ex vivo. The rs13900 alleles imparted differential activity to reporter vectors and influenced the translatability of the reporter

transcript. We further demonstrated the role of HuR in mediating allele-specific effects on CCL2 expression in overexpression and silencing studies. Our studies suggest that the differential interactions of HuR with rs13900 could modulate CCL2 expression and could in part explain the interindividual differences in CCL2-mediated disease susceptibility.

## Introduction

Identification of functional and/or causal genetic variants continues to pose a significant challenge in the post-genome-wide association studies (post-GWASs) era (*MacArthur et al., 2017*; *Visscher et al., 2017*; *Tam et al., 2019*). While emphasis has been placed on polymorphisms that map to *cis*-elements in enhancers and promoters, genetic variants that disrupt cis-elements in RNA-binding protein (RBP) motifs in 3′UTR have received much less attention. For many genes, the interactions of their 3′ UTRs with specific stabilizing and destabilizing RBPs play a critical role in modulating post-transcriptional events such as mRNA turnover and translatability (*Schwerk and Savan, 2015*; *Mayr, 2019*). The lack of a mechanistic understanding of how single nucleotide polymorphisms (SNPs) localizing to RBP motifs could impact gene expression impedes a greater understanding of the variability in disease susceptibility and outcomes.

Post-transcriptional mechanisms are thought to play a crucial role in the initiation and resolution of the inflammatory response (*Anderson, 2010*; *Yoshinaga and Takeuchi, 2019*). Such regulation can profoundly affect gene expression levels; modest changes in mRNA stability can lead to significant effects on mRNA and protein abundance (*Ross, 1995*; *Buccitelli and Selbach, 2020*). Notably, a genome-scale study using mouse dendritic cells demonstrated that post-transcriptional mRNA degradation was a salient feature of inflammatory and immune signaling genes, as well as targets of NF-kappa B signaling following lipopolysaccharide (LPS) stimulation (*Rabani et al., 2011*). In addition, polymorphisms in non-coding regions such as the 3′ UTR can have a significant impact on mRNA stability and translatability. For example, a genome-wide study of variation in gene-specific mRNA decays in lymphoblastoid cell lines across individuals found about 195 genetic variants that are specifically associated with variation in mRNA decay rates, called 'rdQTLs' (RNA Decay Quantitative Trait Loci) (*Pai et al., 2012*). The authors estimated that 35% of the most significant expression quantitative trait loci (eQTLs) SNPs are associated with decay rates (*Pai et al., 2012*). In another study, Duan et al. reported that ~37% of gene expression differences among individuals may be attributed to RNA half-life differences (*Duan et al., 2013*). Farh et al. reported that the 3′ UTRs are highly enriched for eQTL candidate causal SNPs (>1500) relative to other transcribed SNPs (*Farh et al., 2015*). However, the molecular mechanisms by which these polymorphisms alter mRNA stability or translatability remain poorly understood.

CCL2 is a potent monocyte chemoattractant produced by various cell types, either constitutively or following activation. CCL2 expression can be regulated by inflammatory molecules (e.g., IL-1, TNFα, LPS, and IFNγ) and growth factors (e.g. PDGF). While the cells of monocyte-macrophage lineage are a major source of CCL2 (*Yoshimura et al., 1989*), other cell types, such as fibroblasts, astrocytes, epithelial, and endothelial cells, are also an important source (*Deshmane et al., 2009*). CCL2 mediates recruitment of monocytes, memory T-cells, and dendritic cells to the site of inflammation (*Melgarejo et al., 2009*; *Gschwandtner et al., 2019*), and there is substantial evidence implicating CCL2 as a key mediator in macrophage-mediated diseases. Rovin et al. described an SNP in the 5′-regulatory region of *CCL2* annotated as rs1024611 dbSNP database; originally designated as –2518A>G (*Rovin et al., 1999*) or –2578A>G (*Gonzalez et al., 2002*) that was associated with increased plasma CCL2 expression (*Rovin et al., 1999*). This SNP is associated with increased serum CCL2 levels, enhanced macrophage recruitment to tissues, and progression to HIV-associated dementia (*Gonzalez et al., 2002*). Other studies showed that the rs1024611 G allele is associated with increased CCL2 levels in the plasma, urine, and cerebrospinal fluid in health and disease, and in tissues such as liver and skin (*Letendre et al., 2004*; *Joven et al., 2006*; *Cho et al., 2004*; *McDermott et al., 2005*; *Fenoglio et al., 2004*). The rs1024611 polymorphism has been associated with several diseases, including myocardial infarction (*McDermott et al., 2005*), carotid atherosclerosis (*Alonso-Villaverde et al., 2004*), pulmonary tuberculosis (*Flores-Villanueva et al., 2005*), severe acute pancreatitis (*Cavestro et al., 2010*), lupus nephritis (*Tucci et al., 2004*), asthma susceptibility and severity (*Szalai et al., 2001*), Crohn's disease (CD) (*Palmieri et al., 2010*), Alzheimer's disease (*Fenoglio et al., 2004*), and

infections by Japanese encephalitis virus (*Chowdhury and Khan, 2017*), and SARS-CoV-1 (*Tu et al., 2015*). Notably, rs3091315, a GWAS risk variant for CD (*Franke et al., 2010*) and inflammatory bowel disease (IBD) (*Liu et al., 2015*), is in a strong linkage disequilibrium (LD) with both rs1024611 (D'=1.0, $r^2$=0.98) and rs13900 (D'=1.0, $r^2$=0.98) in the CEU population. Given the importance of disease associations with rs1024611, significant efforts have been made by other groups and us to understand the molecular basis of this differential CCL2 expression associated with this polymorphism (*Gonzalez et al., 2002*; *Mummidi et al., 2009*; *Wright et al., 2008*; *Page et al., 2011*; *Pham et al., 2012*). However, these studies did not provide a mechanistic link between the rs1024611 polymorphism and CCL2 expression, giving rise to the possibility that SNPs in strong LD with rs1024611 could be mediating these effects.

To identify potential functional SNPs that could explain the variability in CCL2 expression, we developed an extensive LD map of the *CCL2* genomic locus and reported that an SNP designated as the rs13900 (NM_002982.4:c.*65=) in the *CCL2* 3′UTR is in perfect LD with rs1024611 and can serve as its proxy (*Pham et al., 2012*). We and others have demonstrated AEI in *CCL2* using rs13900 as a marker with the T allele showing a higher expression level relative to C allele (*Johnson et al., 2008*; *Pham et al., 2012*). In this study, we show that the differential binding of Human Antigen R (HuR), an RBP previously implicated in CCL2 expression, leads to altered stability and translatability of *CCL2* transcripts, providing a mechanistic explanation for increased CCL2 expression in individuals with the rs1024611G-rs13900T haplotype and inter-individual differences in disease susceptibility associated with this haplotype.

## Results

### Individuals heterozygous for rs13900 show AEI of *CCL2*

We and others have previously reported a perfect LD between rs1024611 in the *CCL2* cis-regulatory region and rs13900 in its 3′ UTR and that rs13900 can serve as a proxy for the disease-associated rs1024611 (*Hubal et al., 2010*; *Intemann et al., 2011*; *Kasztelewicz et al., 2017*; *Pham et al., 2012*). We further showed that *CCL2* exhibits allelic expression imbalance (AEI) in heterozygous individuals, with the rs13900 T allele (alternative allele) having a higher expression than the rs13900 C allele (reference allele) (*Figure 1*). For this study, we recruited 47 healthy unrelated individuals (18–35 years of age) who were screened for rs13900 (*Figure 1—figure supplement 1*). *Supplementary file 1* shows the genotype and allele frequencies for rs13900 polymorphism in these recruited individuals. We found that the rs13900 C allele was at a higher frequency than the rs13900 T allele, and the genotype frequencies were in line with Hardy-Weinberg equilibrium (p≥0.05). We reconfirmed that *CCL2* exhibits AEI using data from heterozygous individuals. For this, we used total RNA obtained from LPS-treated peripheral blood mononuclear cell (PBMC) as described previously (*Pham et al., 2012*). LPS induced *CCL2* expression in PBMCs as confirmed by real-time quantitative PCR (RT-qPCR), with about a 4.3-fold increase at 1 hr, 6.09-fold increase at 3 hr, and 1.94-fold increase at 6 hr (*Figure 1—figure supplement 2*). We used the 3 hr stimulation in the subsequent experiments as the peak *CCL2* expression was detected at this time point following LPS stimulation. AEI was measured by quantifying the relative amount of the two alleles, i.e., alternative allele (T) to reference allele (C), measured from the chromatogram after normalization of peak intensity using PeakPeaker v.2.0 (*Figure 1B and C*). This strategy ensures a direct comparison between the amount of *CCL2* mRNA that is transcribed from each allele or haplotype and that each allele is equally subjected to the effects of any external factors. gDNA was utilized as a control. The boxplot illustrates a notable difference in the detected levels of the C and T allele in cDNA with a higher expression of T allele relative to C allele (p<0.005) (*Figure 1D*).

### *CCL2* mRNA transcripts bearing rs13900 C and T alleles have different stability

Previous studies have shown that *CCL2* mRNA is subjected to post-transcriptional regulation through modulation of mRNA stability (*Hao and Baltimore, 2009*). Here, we determined whether the rs13900 modulates mRNA stability, which may in part explain AEI. We used purified monocytes from heterozygous individuals that were left untreated or stimulated with LPS for 3 hr. Cells were either harvested after 3 hr of LPS stimulation (considered as t=0) or cultured in the presence or absence (DMSO) of

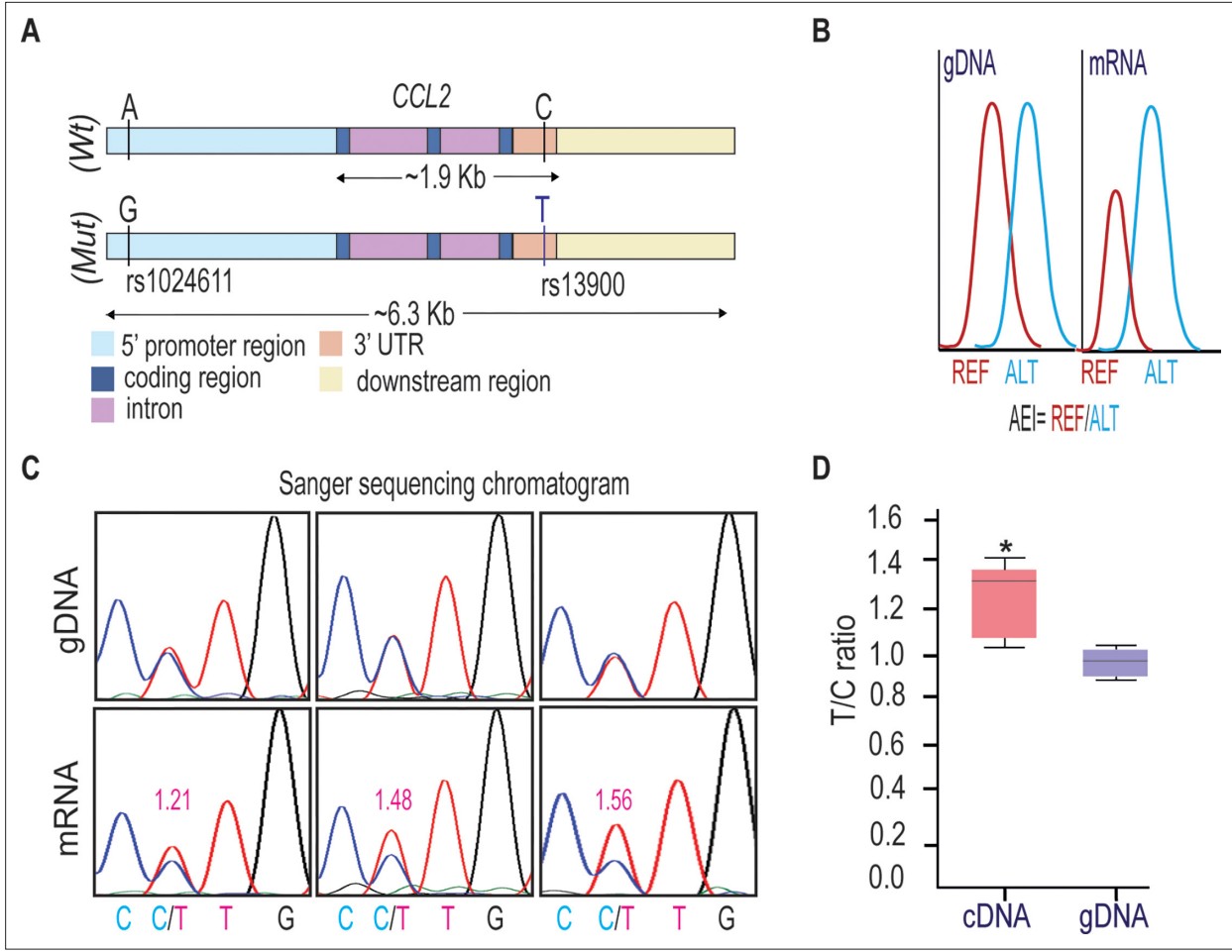

**Figure 1.** rs13900 heterozygous individuals exhibit allelic expression imbalance (AEI) in *CCL2*. (**A**) Schematic depicting distal, proximal regulatory elements extending 3 kb on either side of *CCL2* gene and the linkage disequilibrium (LD) between regulatory polymorphism rs1024611 and the transcribed polymorphism rs13900. rs1024611 is located 2578 base pairs upstream of the *CCL2* translation start site and rs13900 is located in the *CCL2* 3'untranslated region (3' UTR). (**B**) Allelic expression imbalance (AEI) in heterozygous donors is measured as a ratio of alternative allele (ALT) to reference allele (REF) in a transcribed polymorphism. (**C**) Representative chromatograms obtained following Sanger sequencing of PCR products obtained from genomic DNA (gDNA) and reverse transcription PCR of mRNA (cDNA) from three individuals heterozygous for rs13900. gDNA and mRNA were obtained from peripheral blood mononuclear cell (PBMC) treated with lipopolysaccharide (LPS) for 3 hr as previously described. The allelic ratios shown were determined by PeakPicker analysis. PeakPicker calculates allelic ratios by dividing the peak height of the alternate allele (rs13900 T allele) by that of reference allele (rs13900 C allele). The gDNA peaks were used for normalization. (**D**) Allelic ratio for cDNA and gDNA in six individuals heterozygous for rs13900 after treatment with LPS for 3 hr. Statistical significance for the difference in the level of expression between the alleles was determined using Student's t test (p<0.003).

The online version of this article includes the following source data and figure supplement(s) for figure 1:

**Source data 1.** Numerical data used to generate *Figure 1D*.

**Figure supplement 1.** Single nucleotide polymorphism (SNP) genotyping for rs13900 using TaqMan technology.

**Figure supplement 1—source data 1.** Numerical data used to generate *Figure 1—figure supplement 2*.

**Figure supplement 2.** Time course of *CCL2* mRNA expression in peripheral blood mononuclear cells (PBMCs).

the transcriptional inhibitor actinomycin D for an additional 1, 2, or 4 hr. Total RNA was isolated at each time point to assess *CCL2* transcript, calculated as fold induction over unstimulated cells. *CCL2* mRNA levels showed a strong upregulation following LPS stimulation (p<0.05) (*Figure 2A*). Actinomycin D treatment revealed the kinetics of *CCL2* transcript degradation by determining the mRNA half-life, which represents the time (expressed in hours) at which mRNA expression is 50% of the initial level (*Figure 2B*; $t_{1/2}=\ln(0.5)/\text{slope}$). For *CCL2*, $t_{1/2}=1.763$ hr and is in line with previously published

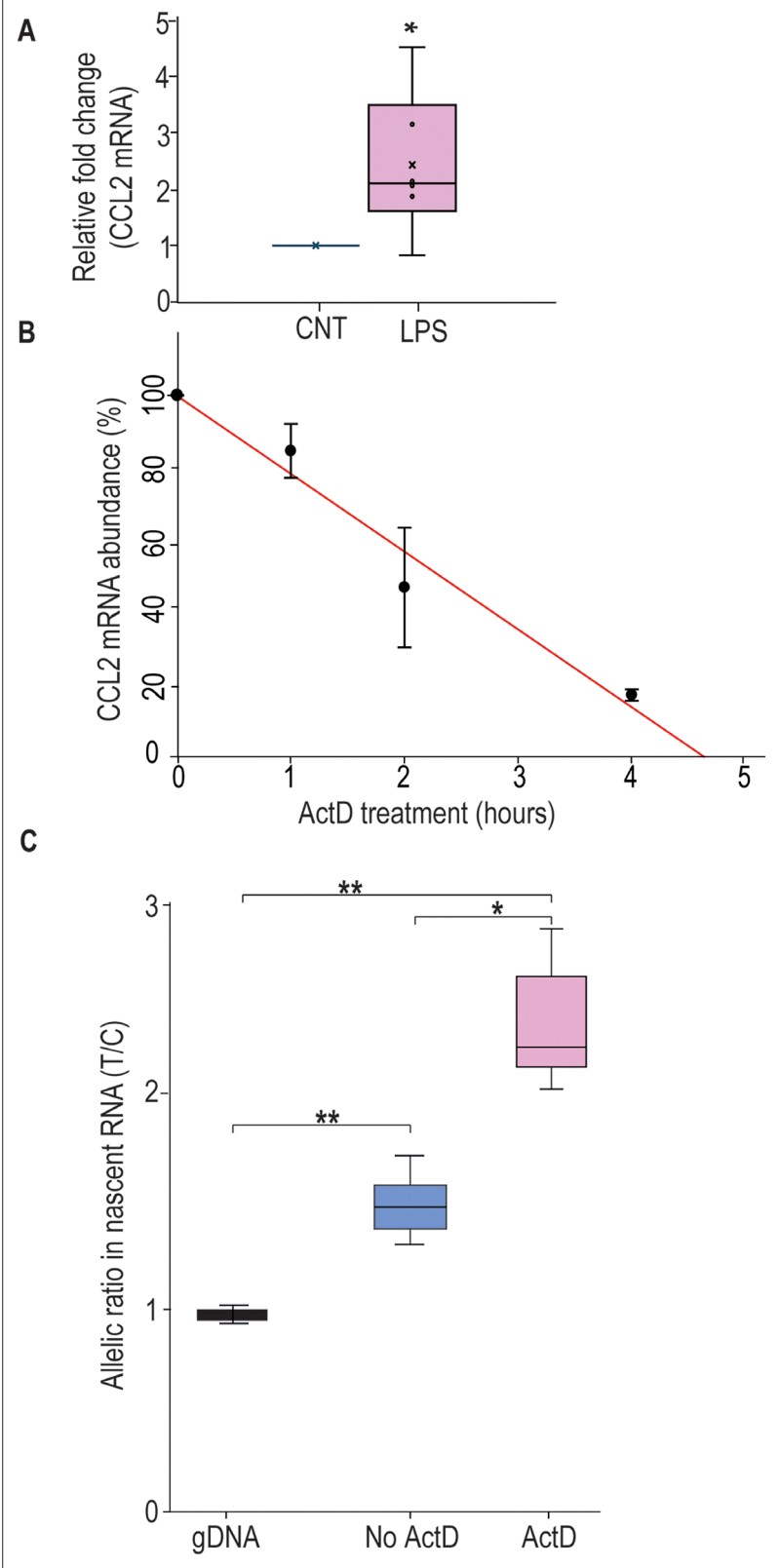

**Figure 2.** rs13900T confers greater stability to *CCL2* mRNA. (**A**) *CCL2* mRNA expression in peripheral monocytes of heterozygous individuals (N=6) after treatment with lipopolysaccharide (LPS) for 3 hr and then incubated with 5 µg actinomycin D (Act D) for indicated times. mRNA was detected by RT-PCR. Results, normalized to 18S rRNA levels, are expressed as fold-increase over unstimulated cells (CNT). Levels shown in the bar graph represent

*Figure 2 continued on next page*

*Figure 2 continued*

mean ± SEM of result at time 0 (*p=0.019) versus unstimulated cells. (**B**) *CCL2* mRNA half-life, calculated for each condition as the time (in hours) required for the transcript to decrease to 50% of its initial abundance ($t_{1/2}$=ln (0.5)/ slope). (**C**) Nascent RNA was isolated from treated monocytes from three individuals in the presence and absence of ActD. Allelic ratio was determined after 4 hr of incubation with or without ActD. Expression of the rs13900 T allele was much higher in ActD-treated samples. The difference between the groups was assessed by ANOVA with Fisher's least significant difference (LSD) method (*p<0.05, **p<0.005).

The online version of this article includes the following source data for figure 2:

**Source data 1.** Numerical data used to generate *Figure 2A and C*.

studies that *CCL2* mRNA stability is modulated following inflammatory and cytokine stimuli (*Hao and Baltimore, 2009*; *Zhai et al., 2008*).

To rule out the confounding effects of preexisting mRNA, the relative stability of rs13900 C- and T-allele bearing transcripts in heterozygous individuals was evaluated using nascent RNA. Using nascent RNA allows for accurate determination of mRNA decay by eliminating the effects of preexisting mRNA. Briefly, monocytes were stimulated with LPS in the presence of 5-ethynyl uridine (EU) for 3 hr. Cells were then washed and further incubated with actinomycin D or DMSO for up to 4 hr. Following the treatment, we assessed mRNA stability according to previously described protocols using macrophages (*Hao and Baltimore, 2009*; *Mahmoud et al., 2014*), and the nascent RNA was captured after 0, 1, 2, or 4 hr using click reaction technique. The click reaction adds a biotin handle to nascent RNA which is then captured by streptavidin beads. cDNA was synthesized from the captured nascent RNA, PCR-amplified, and expression of the individual alleles was assessed as described above. While the allelic ratio in the DMSO control samples was ~1.6 in accordance with our initial observation, the allelic ratio was further increased in the presence of actinomycin D (*Figure 2C*). Performing these experiments in heterozygous donors obviates any concerns regarding the actinomycin D-induced cellular toxicity as it is likely to have similar effects on *CCL2* transcripts with either rs13900 C or T. Overall, our results suggest that post-transcriptional mechanisms such as RNA stability may be playing a role in rs13900-mediated CCL2 AEI.

## Bioinformatic analyses of rs13900

While previous experimental studies showed that *CCL2* 3' UTR binds HuR, it is not known whether rs13900 disrupts or alters HuR binding (*Lebedeva et al., 2011*; *Fan et al., 2011*). Therefore, we analyzed the rs13900 flanking region using various bioinformatic software to mine existing whole-genome datasets (e.g. PAR-CLIP datasets) and to predict any mRNA structural changes and altered RBP motifs (*Supplementary file 2*). We used AURA to examine the colocalization between rs13900 and HuR (*Figure 3A*). As genetic variants can also alter mRNA secondary structure, we used the ViennaRNA Package to assess changes in the *CCL2* secondary structure due to rs13900. As shown in *Figure 3B*, the rs13900 T allele could potentially alter the *CCL2* transcript secondary structure (black arrow). Further bioinformatic analysis was performed using the POSTAR3 suite which incorporates HOMER (*Heinz et al., 2010*) for motif analysis and RNA context (*Kazan et al., 2010*) that identifies not only known but also predicts relative binding and structural preferences of RBPs. HOMER motif analysis (*Heinz et al., 2010*) identified a HuR-binding motif in the region flanking the rs13900 (*Figure 3C*). *Figure 3D* shows the relative structural preference of HuR to different structural contexts identified by RNAcontext, where the letters P, L, U, M indicate that the nucleotide is paired (P), in a hairpin loop (L), in an unstructured (or external) region (U), or miscellaneous (M). The M category includes various unpaired contexts such as nucleotide localizing to a bulge, internal loop, or multiloop. The rs13900 C allele to rs13900 T allele transition is predicted to form a stem (*Figure 3B*), which is predicted to increase HuR binding. We also used RBP-Var, a bioinformatics tool that scores variants involved in post-transcriptional interaction and regulation, which predicted a score of 1e for rs13900 (*Mao et al., 2016*). This annotation system rates the functional confidence of variants from category 1 to 6. While category 1 is the most significant category and includes variants that are known to be eQTLs, likely affecting RBP-binding site, RNA secondary structure, and expression, category 6 is assigned to minimal possibility to affect RBP binding. Additionally, subcategories provide further annotation ranging from the most informational variant (a) to the least informational variant (e). The reported 1e score denotes that the variant has a motif for RBP binding. While the employed scoring

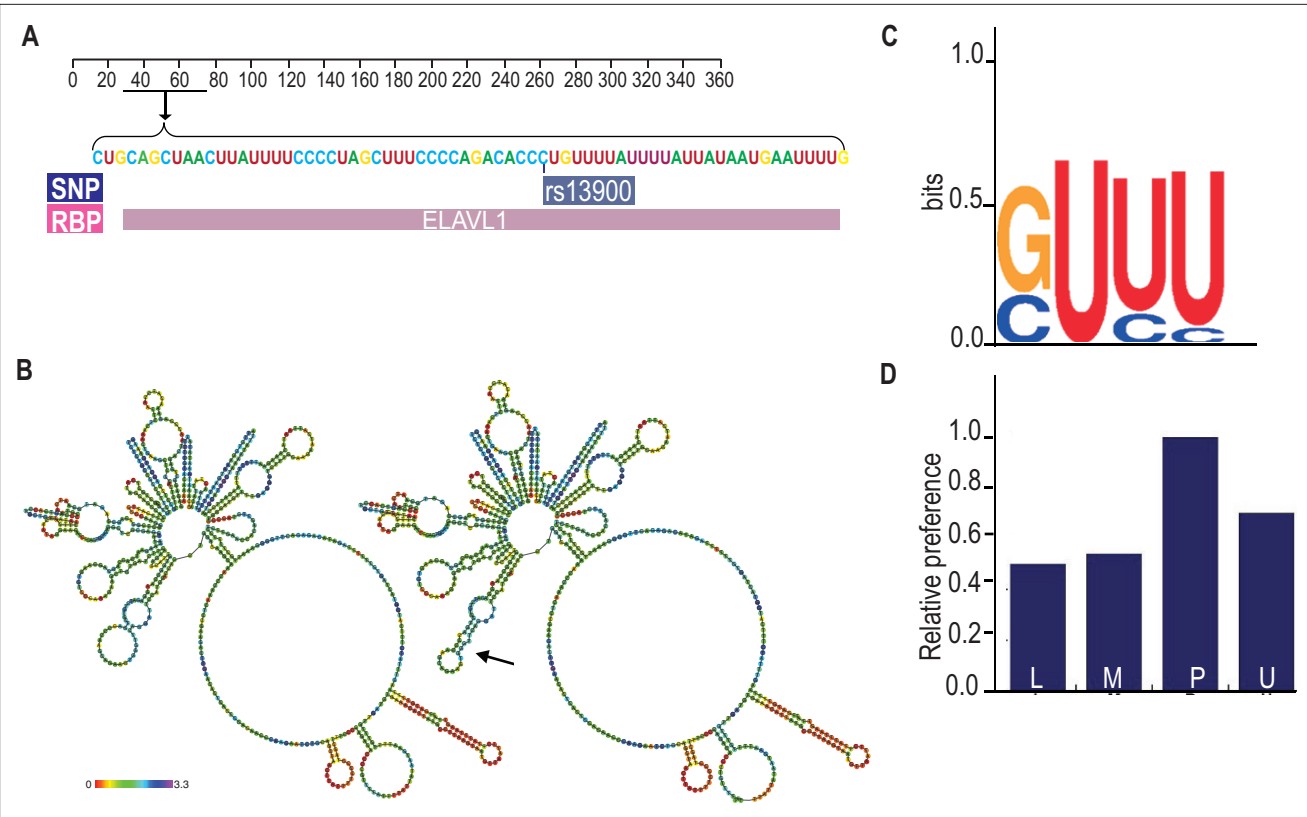

**Figure 3.** Bioinformatic analysis of the rs13900. (**A**) Validation of RBP-binding sites and polymorphism located on the 3' untranslated region (3' UTR) of *CCL2* transcript from the Atlas of UTR Regulatory Activity and analysis of ENCODE genome-wide datasets detected specific enrichment of HuR (ELAVL1) at the region that contains the rs13900. (**B**) Predicted changes in the secondary structure using ViennaRNA Package 2.0; the black arrow indicates the changes in the secondary structure due to the rs13900 T allele. (**C**) Sequence logo of HuR-binding site as determined by HOMER. (**D**) Relative structural preference of HuR across different nucleotide contexts: P denotes paired regions, L denotes hairpin loops, U denotes unstructured (or external) regions, and M denotes miscellaneous regions.

system is hierarchical from 1a to 1e, with decreasing confidence in the variant's function, the variants in category 1 are considered potentially functional to some degree. Taken together, our bioinformatic analysis suggested that the rs13900 allele could potentially alter the binding of HuR to *CCL2* 3' UTR and may have functional consequences.

## Differential binding of HuR to rs13900 C and T alleles in vitro

To experimentally verify our bioinformatic findings on rs13900, we utilized RNA electrophoretic mobility shift assay (REMSA) to determine whether the region of CCL2 3' UTR that flanks rs13900 binds in vitro to HuR and if there are allelic differences in binding. Purified labeled single-stranded oligoribonucleotides corresponding to either rs13900 C or rs13900 T alleles were incubated with 10 µg of whole-cell extracts. We tested whether the bound complexes contained HuR by performing antibody-mediated supershift assays. As shown in *Figure 4A and B*, a predominant shift was observed for the oligoribonucleotide corresponding to rs13900 T allele (lane 8). Notably, there was an approximately sevenfold difference (p<0.005) in oligoribonucleotide/HuR/antibody complexes generated with oligoribonucleotide corresponding to rs13900 T allele when compared to those generated with rs13900 C allele (lane 4). The specificity of the complex formation was confirmed by using a nonspecific antibody (*Figure 4A,* lanes 3 and 7). To rule out the possibility that additional RBPs may be involved in the complex formation with whole cell extracts, we used purified recombinant HuR protein in mobility shift assays. We confirmed that a higher percentage of the complex formation is seen with rs 13900 T oligoribonucleotide when compared with rs 13900 C oligoribonucleotide even with the purified protein, suggesting that C to T transition leads to increased binding affinity (*Figure 4C and D*; *Figure 4—figure supplement 1*).

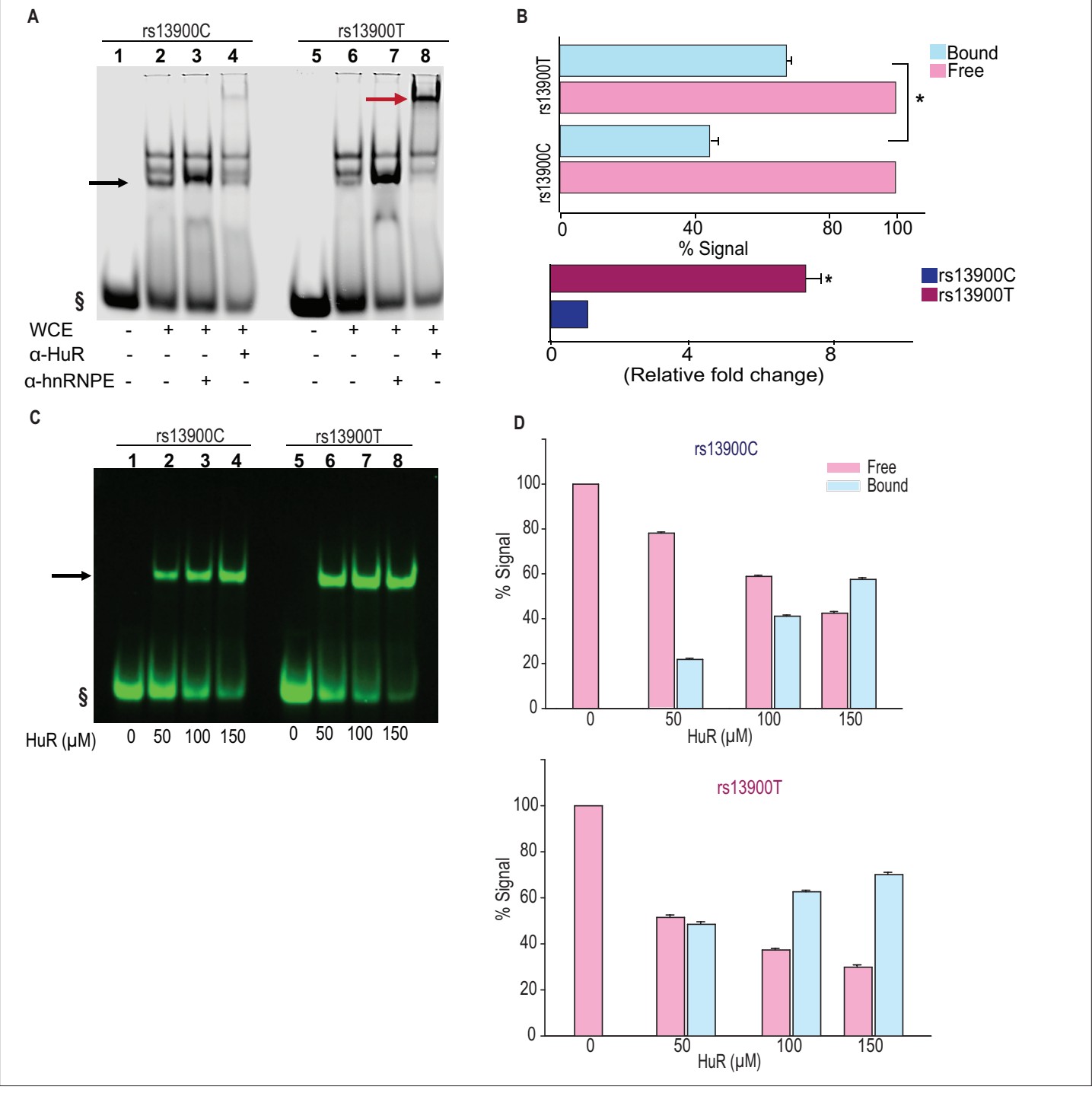

**Figure 4.** The rs13900 T allele shows increased in vitro binding of HuR. (**A**) RNA electrophoretic mobility shift assay (REMSA) with labeled oligoribonucleotide containing either rs13900 C or T allele and whole cell extracts from HEK-293 cells. § denotes free probe; black arrow, bound probe; red arrow, supershift. (**B**) Representative quantitative densitometric analysis of the antibody-shifted complexes suggested increased HuR binding to the oligoribonucleotide bearing rs13900 T allele. The signals in the bound fraction(s) were normalized using the free probe. The top panel represents the data from three independent experiments (mean ± SEM). Statistical analyses were performed using Student's t test (*p<0.001). The bottom panel shows the relative fold enrichment of the bound protein complexes to the oligoribonucleotide containing the rs13900 T allele relative to that containing the rs13900 C allele. Statistical significance was calculated using Student's t test (*p<0.001). (**C**) REMSA with labeled oligoribonucleotides containing either rs13900 T or C allele and purified recombinant HuR protein at indicated concentrations. § denotes free probe; black arrow, bound probe. (**D**) Plot showing the fraction of bound rs13900 C or rs13900 T oligoribonucleotides with increasing HuR concentrations. The signal in the bound fractions was normalized with free probe. The figure represents data from three independent experiments (mean ± SEM).

*Figure 4 continued on next page*

*Figure 4 continued*

The online version of this article includes the following source data and figure supplement(s) for figure 4:

**Source data 1.** PDF file containing original gel shift assay blots for *Figure 4A and C*, indicating the relevant band and treatments.

**Source data 2.** Original files for gel shift assay analysis displayed in *Figure 4A and C*.

**Source data 3.** Numerical data used to generate *Figure 4B* (upper and lower panels).

**Figure supplement 1.** Differential binding of rs13900 C or rs13900 T allele with recombinant HuR protein.

**Figure supplement 1—source data 1.** PDF file containing original gel shift assay blot for *Figure 4—figure supplement 1*, indicating the relevant band and treatments.

**Figure supplement 1—source data 2.** Original gel shift assay blots displayed in *Figure 4—figure supplement 1*.

## Differential binding of HuR to rs13900 C and T alleles ex vivo

We next tested the hypothesis that rs13900 is associated with altered binding affinity to HuR ex vivo. For this, we performed RNA immunoprecipitation (RIP) in monocyte/macrophages derived from four heterozygous individuals (*Figure 5—figure supplement 1*). Immunoprecipitations were performed using cytoplasmic lysates of macrophages treated with LPS using an affinity-purified HuR antibody or IgG. *Figure 5A* shows the relative enrichment of HuR in the immunoprecipitated fraction compared to the IgG control. The HuR immunoprecipitated fraction showed significant enrichment ($p<0.05$) of the region encompassing rs13900 when compared to the IgG control (*Figure 5B and C*). *The CCL2* transcript was enriched ~10-fold in the HuR immunoprecipitated fraction in comparison to the IgG-bound fraction (*Figure 5C*). Notably, transcripts corresponding to the rs13900 T allele were enriched relative to transcripts containing the rs13900 C allele ($p<0.05$) in the anti-HuR RNA immunoprecipitation (RIP) complexes (*Figure 5D*).

## rs13900 T allele confers increased mRNA stability in reporter assays

As HuR is implicated in mRNA stability of many mRNA transcripts, including *CCL2*, we tested the hypothesis that *CCL2* 3′ UTR influences its stability and that rs13900 modulates this effect. We constructed reporter plasmids that harbor the *CCL2* 3′ UTR containing either rs13900 C or rs13900 T allele (*Figure 6A*) and nucleofected them into HEK-293 cells as described in Materials and methods. Luciferase activity was measured after 24 hr post-transfection (*Figure 6B*). Our results indicated that the presence of *CCL2* 3′ UTR significantly ($p<0.05$) reduced the luciferase activity. However, cells transfected with plasmids bearing the rs13900 T allele showed higher luciferase activity when compared with cells transfected with plasmids bearing the rs13900 C allele, suggesting that the presence of rs13900 T allele conferred increased stability to the transcript ($p<0.05$). We next analyzed the influence of HuR overexpression in reporter vectors containing *CCL2* 3′ UTR with either rs13900 C or T alleles. Overexpression of HuR led to a significant increase in luciferase activity of the reporter vector bearing rs13900 T allele ($p<0.05$). However, HuR overexpression had no significant effect on the luciferase activity of the reporter vector bearing the rs13900 C allele (*Figure 6C*). Conversely, we examined the effect of HuR on the expression of the reporter assay by co-transfection of HuR siRNA and luciferase reporter constructs. While HuR knockdown had no effect on luciferase activity of the reporter construct bearing the rs13900 C allele, it caused a significant reduction in luciferase activity of the construct bearing the rs13900 T allele ($p<0.05$) (*Figure 6D*). Overexpression and knockdown of HuR were confirmed by using Western blots (*Figure 6—figure supplement 1*).

## Role of HuR-rs13900 interactions in *CCL2* mRNA translatability

The differential interactions with HuR by rs13900 C and rs13900 T alleles could potentially alter *CCL2* mRNA translatability as HuR could increase mRNA translation (*Schultz et al., 2020*). Therefore, we assessed the relative allelic enrichment in the monosomal and polysomal fractions obtained from monocyte-derived macrophages (MDMs) from two individuals exhibiting AEI. Translationally active and inactive pools of RNA were fractionated by isolating the monosomal and polysomal fractions from MDMs on sucrose gradients after the cells were treated with cycloheximide to block translation. The distribution of *CCL2* mRNA in macrophages cultured in the presence or absence of LPS for 3 hr is depicted in *Figure 7—figure supplement 1*. As previously reported by others, LPS treatment resulted in a distinct shift of the mRNAs from the monosomal fraction to the polysomal fraction (*Schott et al.,*

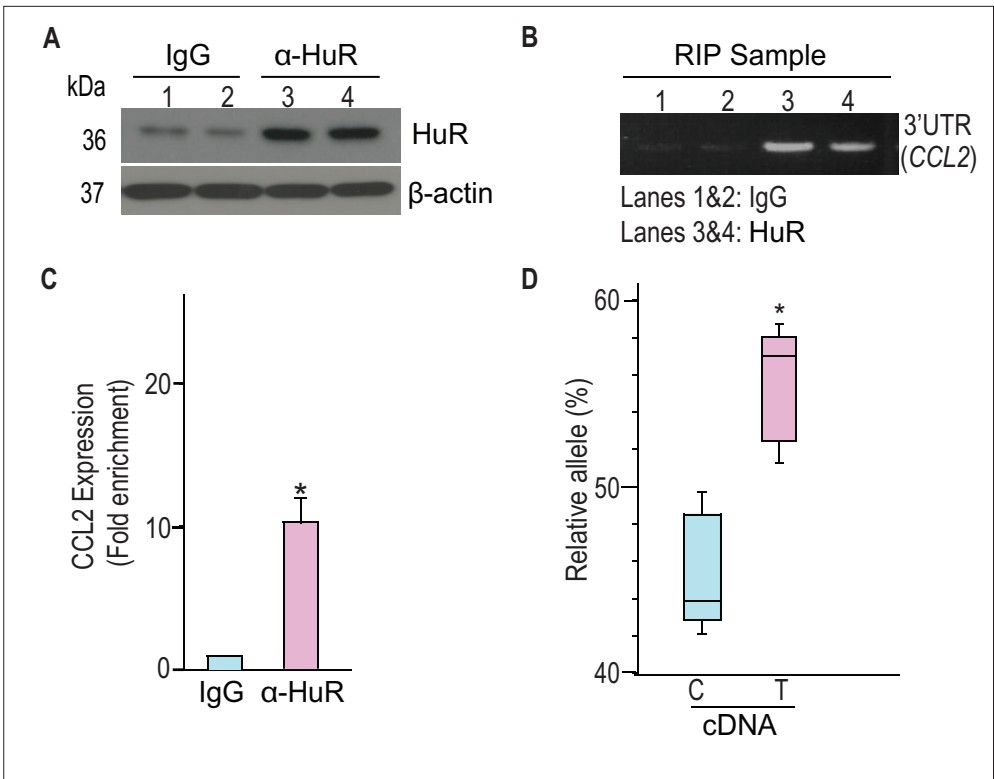

**Figure 5.** rs13900 C and T alleles are associated with differential binding to HuR ex vivo. (**A**) HuR enrichment in immunoprecipitated material from macrophages stimulated with lipopolysaccharide (LPS). The input sample could not be included due to limited availability of material. To ensure comparable protein recovery across samples, β-actin was used as a loading control. (**B**) *CCL2* 3' untranslated region (3' UTR) was detected at significant levels in the samples precipitated by anti-HuR antibody when compared to the control IgG. (**C**) *CCL2* mRNA expression in anti-HuR antibody enriched immunoprecipitated material analyzed by real-time quantitative PCR (RT-qPCR) (N=4). Statistical significance was calculated using Student's t test (*p<0.005). The error bars represent SEM. (**D**) Relative expression levels of rs13900 C and T alleles in the anti-HuR-enriched immunoprecipitated complexes obtained from macrophages stimulated with LPS (N=6). Statistical significance was calculated using Student's t test (*p<0.005).

The online version of this article includes the following source data and figure supplement(s) for figure 5:

**Source data 1.** PDF file containing original western blots for supporting *Figure 5A*.

**Source data 2.** Original files for western blot analysis displayed in *Figure 5—source data 1*.

**Source data 3.** PDF file containing original uncropped gel for *Figure 5B*, indicating the relevant bands and treatment.

**Source data 4.** Original files for agarose gel displayed in *Figure 5B*.

**Source data 5.** Numerical data used to generate *Figure 5C and D*.

**Figure supplement 1.** Flow cytometric analysis of cell surface markers to assess the purity of monocytes isolated from peripheral blood mononuclear cell (PBMC) and in vitro differentiation to macrophages.

*2014*). Consistent with this prior report, 60.4% of the *CCL2* mRNA was associated with polysomal fraction following stimulation of cells with LPS. We assessed the differences in allelic enrichment in cytosolic, monosomal, and polysomic fractions and found that the polysomal fractions showed the enrichment of the rs13900 T allele (*Supplementary file 3*). However, this differential loading of the rs13900 T allele was noted for one donor in cytosolic and monosomal fractions as well.

To further address this question, we used a reporter-based system to assess the effect of rs13900 C and T alleles on translatability as previously reported (*Zhang et al., 2017*). To measure translatability, luciferase mRNA and protein were measured simultaneously, and translatability was calculated as luciferase activity normalized by the luciferase mRNA levels after adjusting for protein and 18S rRNA

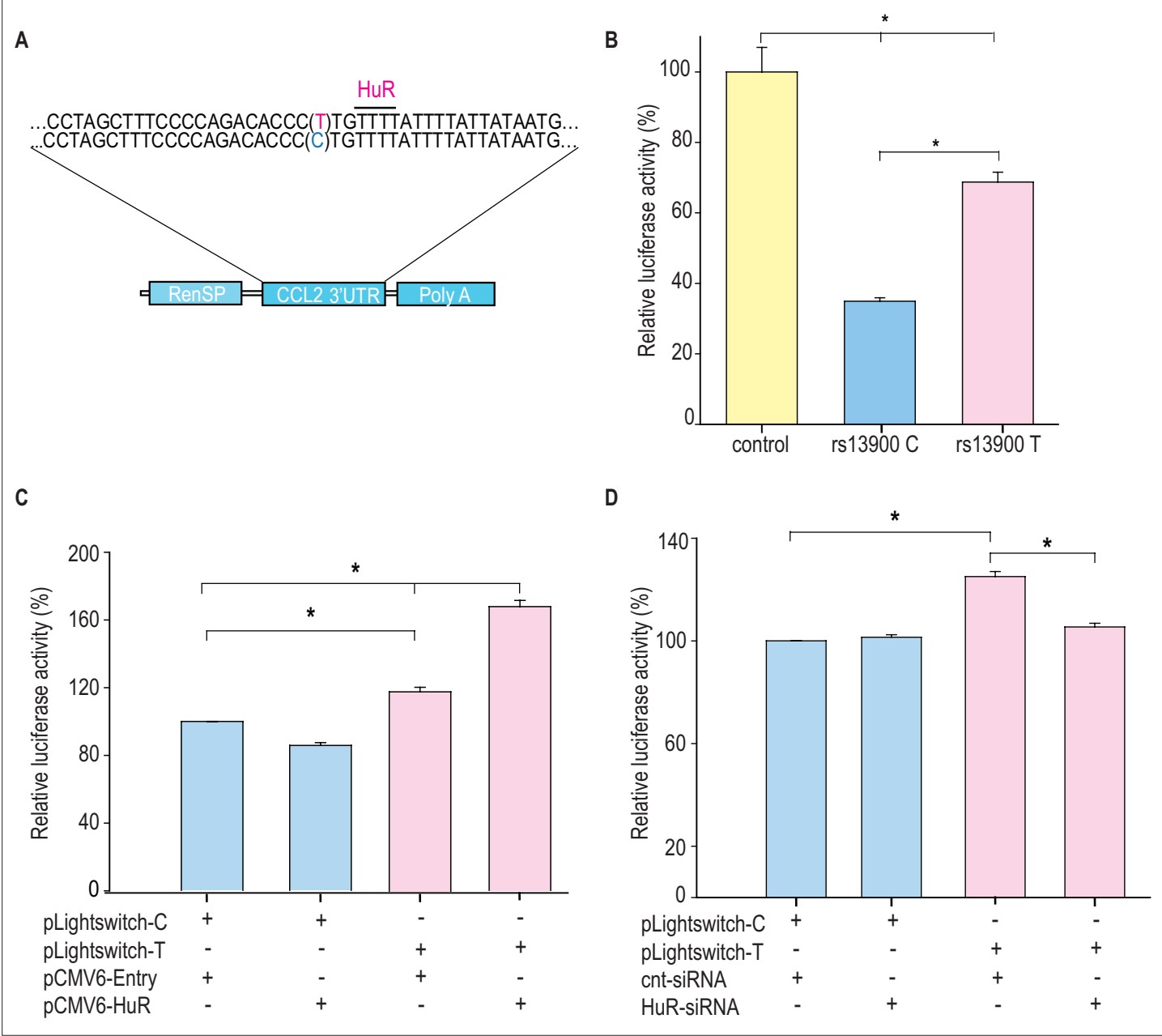

**Figure 6.** Differential effects of rs13900 alleles in reporter assays and role of HuR. (**A**) Schematic representation of the luciferase reporter vectors containing *CCL2* 3'untranslated region (3' UTR) with either rs13900 C or T allele. (**B**) HEK-293 cells were transfected with the equal quantities of *CCL2* 3' UTR reporter vectors, and luciferase activity was measured 48 hr later. The relative luciferase activities of the 3' UTR reporter plasmids were expressed as a percentage reduction in the luminescence when compared to the control vector that was set to 100% after normalizing for the protein content of the lysates. The error bars indicate the standard error of mean, and statistical significance was calculated using two-tailed Student's t test (*p<0.05) (N=3). (**C**) HEK-293 cells were transfected with either pCMV6-HuR (0.5 μg) or pCMV-Entry (0.5 μg), and after 72 hr they were co-transfected with the two plasmid constructs (0.5 μg). Twenty-four hours after transfection, the relative change in luciferase activity was determined (N=3). (**D**) Cells were co-transfected with 125 pmol HuR siRNA or control siRNA and with the two plasmid constructs (0.5 μg). Twenty-four hours after transfection, the relative change in luciferase activity was determined, normalized to total protein concentration, data from three independent experiments (mean ± SEM; N=3). Statistical analyses were performed using Fisher's least significant difference (LSD) method (*p<0.05).

The online version of this article includes the following source data and figure supplement(s) for figure 6:

**Source data 1.** Numerical data used to generate *Figure 6B, C, and D*.

**Figure supplement 1.** Validation of HuR overexpression and silencing by western blotting.

**Figure supplement 1—source data 1.** PDF file containing original western blot for *Figure 6—figure supplement 1*, indicating the relevant band and

*Figure 6 continued on next page*

*Figure 6 continued*

treatments.

**Figure supplement 1—source data 2.** Original files for western blot for *Figure 6—figure supplement 1*.

(*Figure 7A–C*). Our results suggest that rs13900 may alter the mRNA translatability in addition to the transcript stability.

## Differential effect of HuR overexpression on the *CCL2* rs13900 T allele

Our in vitro and ex vivo data indicate that HuR positively regulates *CCL2* transcript stability by increased binding to the T allele. To determine a direct functional relationship between HuR and *CCL2* AEI, we used a lentiviral overexpression system. We either transduced HuR-specific (pCMV6-HuR) or nonspecific control shRNAs (CMV-null) into the monocytes obtained from donors who were either homozygous for the rs13900 C or T allele. Ready to use GFP-tagged pCMV6-HuR or CMV-null lentiviral particles were transduced into macrophages in the presence of polybrene at an MOI of 1 (*Figure 8—figure supplement 1*). Cells were processed 72 hr following virus addition for the analysis of HuR and *CCL2* mRNA expression. Using this lentiviral system, both high transduction efficiency (90%) and high expression levels were achieved in primary human macrophages (*Figure 8—figure supplement 2*). Compared with CMV-null particles, substantial overexpression of HuR was noted with pCMV6-HuR lentiviral particles (*Figure 8A*). HuR overexpression was associated with a higher expression of *CCL2* mRNA in persons homozygous for the T allele relative to those who were homozygous for the C allele (*Figure 8B*).

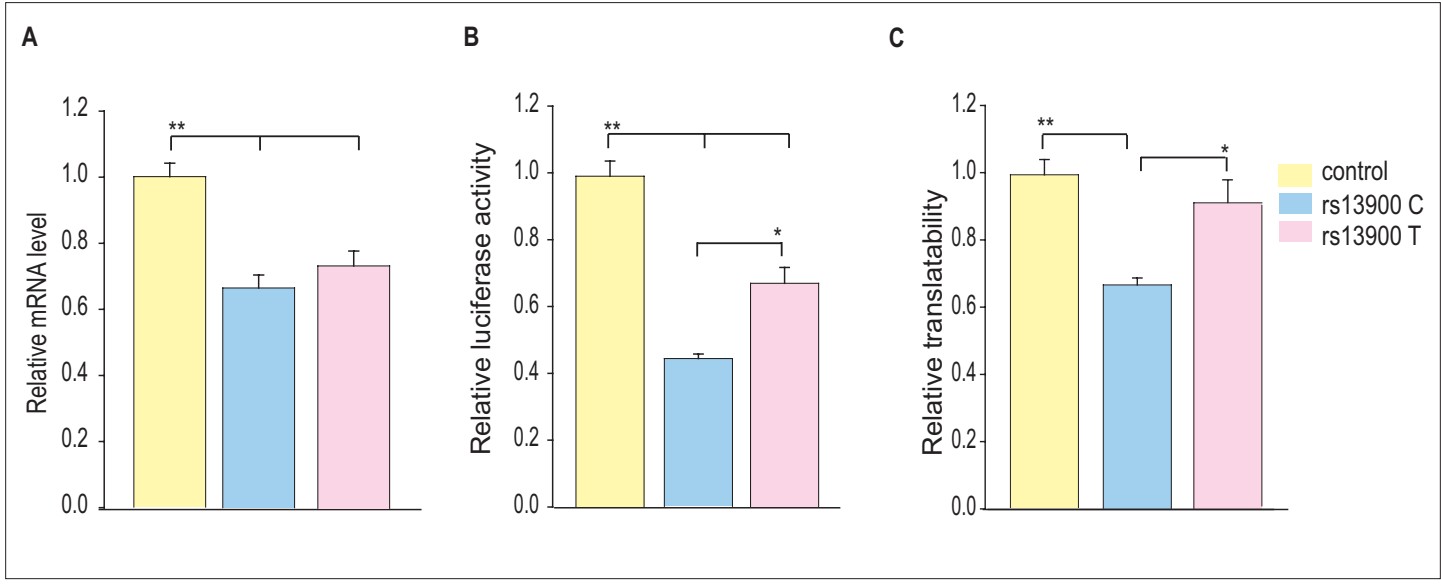

**Figure 7.** Influence of rs13900 C and T alleles on *CCL2* translatability. (**A**) HEK-293 cells were transfected by nucleofection with control and *CCL2* 3' untranslated region (3' UTR) reporter constructs with rs13900 C and T alleles. The nucleofected cells were plated separately and harvested for total RNA isolation or lysed for mRNA level or protein level expression of luciferase, respectively, after 24 hr. The reporter mRNA levels from the transfected 293T cells were quantified by real-time quantitative PCR (RT-qPCR), and 18S rRNA was used for normalization (N=6). (**B**) The relative luciferase activities of the 3' UTR reporter plasmids were expressed as a percentage reduction in the luminescence when compared to the control vector that was set to 100% after normalizing for the protein content of the lysates (N=4). (**C**) mRNA translatability was calculated as luciferase activity normalized by the reporter luciferase mRNA level. The error bars indicate the standard error of mean from four independent experiments (N=4), and statistical significance was calculated using ANOVA and post hoc contrast with Fisher's least significant difference (LSD) method. *p<0.01, **p<0.005.

The online version of this article includes the following source data and figure supplement(s) for figure 7:

**Source data 1.** Numerical data used to generate *Figure 7A, B, and C*.

**Figure supplement 1.** Polysomal association of *CCL2* mRNA before and after lipopolysaccharide (LPS) stimulation of macrophages.

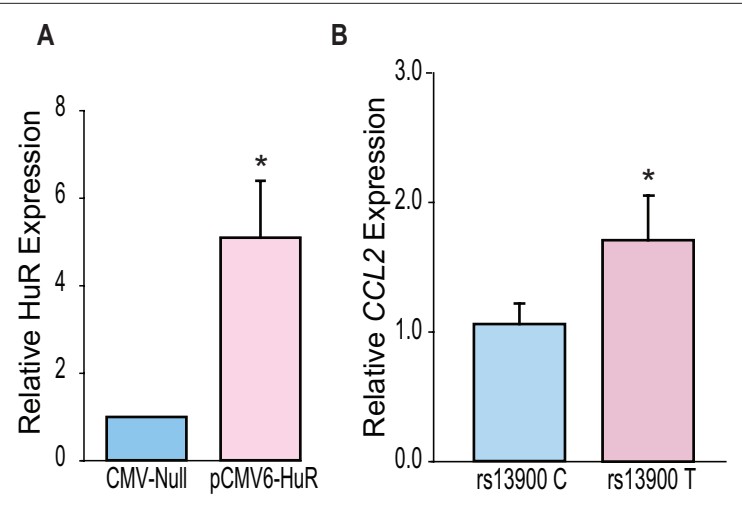

**Figure 8.** HuR differentially regulates *CCL2* haplotypes. (**A**) HuR expression in primary human macrophages following lentiviral transduction. Macrophages obtained from four individuals who are homozygous for either rs13900 C or rs13900 T allele were transduced with either CMV-null or HuR expressing lentiviral particles (pCMV6-HuR) for 72 hr followed by lipopolysaccharide (LPS) stimulation for 3 hr. (**B**) *CCL2* expression determined by real-time quantitative PCR (RT-qPCR) (N=4). Error bars represent SEM. Statistical analyses were performed using Student's t test (*p<0.005).

The online version of this article includes the following source data and figure supplement(s) for figure 8:

**Source data 1.** Numerical data used to generate *Figure 8A and B*.

**Figure supplement 1.** Transduction efficiency of pCMV6-HuR.

**Figure supplement 2.** Lentiviral transduction of differentiated macrophages.

## Discussion

GWASs have led to the discovery of numerous disease susceptibility genetic variants/genes and biological pathways involved in specific diseases, including many with immunological etiologies (*Buniello et al., 2019*; *Visscher et al., 2017*; *Manolio, 2010*). Most of the disease-associated genetic variants identified are present in the non-coding regions of the genome (*Zhang and Lupski, 2015*; *Maurano et al., 2012*; *Zou et al., 2019*). However, characterizing the functional consequence of these genetic variants in the regulatory regions remains a significant challenge to date. About 3.7% of the non-coding variants localize to the untranslated regions (*Steri et al., 2018*), suggesting post-transcriptional mechanisms such as mRNA stability and translatability could determine disease susceptibility (*Maurano et al., 2012*).

The *CCL2* locus exemplifies the difficulty in dissecting the functional consequences of cis-regulatory and non-coding genetic variants. Our analysis of regulatory elements in an ~20 kb region upstream of the human *CCL2* coding region identified highly conserved enhancers and other regulatory elements in the *CCL2* locus (*Bonello et al., 2011*). We identified SNPs with strong LD in this 'super-enhancer' and determined their impact on transcriptional activity, which was minimal (*Pham et al., 2012*). For example, we tested the role of rs7210316 and rs9889296, which had an $r^2>0.9$ with rs1024611 in CEU population and were located ~6.3 kb and ~9.2 kb upstream of it (*Pham et al., 2012*). The genetic regions that encompass and flank these two SNPs either had no or minimal transcriptional activity and lacked the epigenetic markers that have been traditionally associated with gene activation. Additionally, conflicting context-dependent results have been obtained for the transcriptional activities associated with reporter vectors containing rs1024611 A and G alleles, which may have been due to the use of different lengths of the *CCL2* cis-regulatory regions employed in the transcriptional assays. To avoid such context-dependent confounding, we used chromatin annotation, a powerful approach to identify functional SNPs, to generate reporter constructs that span a 6 kb cis-regulatory region (*Pham et al., 2012*). This approach allowed us to directly compare the roles of four correlated SNPs (rs1860190, rs2857654, rs1024611, and rs2857656) on *CCL2* transcriptional activity. We found no significant differences in transcription strength between these two constructs when

transfected into primary human astrocytes (*Pham et al., 2012*). Furthermore, to understand the basis for increased CCL2 expression associated with the rs1024611 G allele, we and others have demonstrated that several transcription factors bind differentially to *CCL2* polymorphisms, including IRF-1, PARP-1, STAT-1, Prep1/Pbx complexes (*Gonzalez et al., 2002*; *Wright et al., 2008*; *Mummidi et al., 2009*; *Page et al., 2011*). However, these previous studies could not unequivocally demonstrate the mechanistic link between differential binding of transcription factors and transcriptional activity and the physiological relevance of implicated transcription factors in CCL2 expression. Our comprehensive studies outlined here demonstrate that the rs13900 T allele differentially binds to the RBP HuR, which could alter CCL2 mRNA stability and gene expression. Furthermore, we provide multiple lines of functional evidence for rs13900/HuR role in *CCL2* AEI, including reporter vector assays and HuR overexpression and depletion experiments.

Our previous finding that rs13900 is in perfect LD with rs1024611 has allowed us to exploit the powerful AEI technique and mechanistically link the *CCL2* rs1024611G-rs13900T haplotype to increased expression (*Pham et al., 2012*). These findings, together with previous knowledge that *CCL2* is subjected to post-transcriptional regulation (*Hao and Baltimore, 2009*) and regulated by RBP (*Fan et al., 2011*), raised the possibility that rs13900 could be functional and impact mRNA stability and translatability. In this study, we determined the half-life of *CCL2* to be 1.76 hr, reinforcing the fact that post-transcriptional regulation is a key feature of *CCL2* expression (*Figure 2B*). Since transcription and RNA degradation are tightly linked, and transcriptional inhibition may lead to mRNA decay, we assessed the allelic differences in expression in nascent RNA obtained from heterozygous individuals. We found that the *CCL2* transcripts with the rs13900 T allele have increased stability relative to those with the rs13900 C allele (*Figure 2C*). In addition, both rs1024611 and rs13900 have suggestive associations with *CCL2* expression in cis-eQTL analyses. For example, rs1024611 and rs13900 are associated with CCL2 expression in MDM infected with *Listeria monocytogenes* (*Nédélec et al., 2016*) ($-\log10(p)=3.19$; effect size $= 0.29 \pm 0.084$) (*Kwong et al., 2021*). Remarkably, rs13900 is also identified as a cis-splicing quantitative trait locus (cis-sQTL) for *CCL2* in both untreated and treated monocytes (*Quach et al., 2016*; *Kwong et al., 2021*; *Alasoo et al., 2018*). Recent evidence suggests that genetic variation influencing RNA splicing could play an important role in determining complex phenotypic traits (*Li et al., 2016*). While rs13900 serves as an sQTL for several *CCL2* transcript variants, we provide an illustrative example here. rs13900 showed significant associations with the canonical *CCL2* transcript, ENST00000225831 in LPS-treated monocytes ($-\log10(p)=9.88$; effect size $= 0.57 \pm 0.084$) and with a CCL2 transcript ENST00000580907 ($-\log10(p)=9.88$; effect size $= -0.57\pm0.084$) that encodes a truncated CCL2 protein (*Kwong et al., 2021*). While the importance of these associations needs more in-depth studies, our observation that *CCL2* transcripts with the rs13900 T allele have a slower degradation could explain the increased CCL2 expression in individuals with the rs1024611G-rs13900 haplotype, as modest changes in mRNA stability can lead to significant effects on mRNA and protein abundance.

Post-transcriptional gene expression is regulated through interactions between the *cis*-elements in mRNAs and their cognate RBPs (*Wu and Brewer, 2012*). Several studies indicate the presence of post-transcriptional operons or regulons – unique subsets of RNAs that are associated with RBPs, which coordinate their localization, translation, and degradation. Our bioinformatics analysis and experimental data confirm HuR binds to the region spanning the rs13900. iClip data from HeLa cells (*Wang et al., 2010*) suggested that the RBPs TIAL1 (T-cell intracellular antigen-1-like protein) and U2AF65 (U2 snRNP auxiliary factor large subunit) could also bind to this region of the *CCL2* transcript (*Mao et al., 2016*). TIAL1 has been proposed to act as a cellular sensor and has been associated with a transcriptome associated with control of inflammation, cell-cell signaling, immune suppression, angiogenesis, metabolism, and cell proliferation (*Reyes et al., 2009*). The role of TIAL1 in *CCL2* expression is not known, and additional studies are needed to determine if its binding contributes to *CCL2* AEI. U2AF65 is a widely expressed splicing factor that associates with RNA polymerase II to bind upstream 3′ splice sites to facilitate splice site pairing in higher eukaryotes (*Hollander et al., 2016*). In addition, a change in RNA structure due to an SNP could indirectly alter accessibility of additional regions to RBPs, and further studies are required to determine any such changes (*Shatoff and Bundschuh, 2020*).

The Hu family contains four members, of which HuR is ubiquitously distributed. HuR consists of three RNA recognition motifs (RRMs) that are highly conserved and canonical in nature (*Ripin et al.,*

*2019*). In the absence of RNA, the three RRMs are flexibly linked, but upon RNA binding, they transition to a more compact arrangement. Mutational analysis revealed that HuR function is inseparably linked to RRM3 dimerization and RNA binding. Dimerization enables recognition of tandem AREs by dimeric HuR (*Pabis et al., 2019*) and explains how this versatile protein family can regulate numerous targets found in pre-mRNAs, mature mRNAs, miRNAs, and long noncoding RNAs. HuR shuttles between the nucleus and cytoplasm and is likely involved in the transport and stabilization of mRNA. HuR action is antagonized by RNA destabilizing proteins such as TTP (tristetraprolin) and AUF (ARE/poly(U)-binding/degradation factor 1) (*Wu and Brewer, 2012*). The role of HuR in chemokine gene regulation has been extensively studied in human airway epithelium cells (*Fan et al., 2011*), and they identified that *CCL2* is one of the top targets for HuR and demonstrated that HuR associated with *CCL2* 3′ UTR in vitro and *CCL2* expression could be modulated by changes in HuR levels using over-expression and RNAi-based experiments. Also, HuR levels influenced *CCL2* mRNA decay. In a mouse model of alcoholic liver disease, HuR was found to play a key role in NOX4-mediated increase in *CCL2* mRNA stability (*Sasaki et al., 2017*). Notably, HuR has also been implicated in mRNA stability of CX3CL1 (*Matsumiya et al., 2010*) and IL-8 (*Choi et al., 2009*) among others. Another study showed that CCL2 itself can induce nuclear to cytoplasmic translocation of HuR and stabilize vascular endothelial growth factor-A (*VEGFA*) mRNA in $CD14^+CD16^{low}$ inflammatory monocytes, thus raising the possibility that CCL2 may stabilize its own message in an autocrine fashion (*Morrison et al., 2014*). These studies further bolster a role for HuR in *CCL2* expression and support our finding that differential binding to HuR alters its expression level.

While there are several examples of genetic variants that influence mRNA stability (*Akdeli et al., 2014*; *Duan et al., 2013*; *Wang et al., 2006*; *Vilmundarson et al., 2021*), very few studies have examined the role of RBPs such as HuR in differentially influencing gene expression levels in humans (*Steri et al., 2018*). A notable exception is the report showing that RBP AUF1 regulates the allele-specific stability of thymidylate synthase (*Pullmann et al., 2006*). Another study showed that a type 2 diabetes-associated polymorphism in the 3′ UTR of *PPP1R3* alters the distance between two ARE motifs and results in differential binding of protein complexes and may be associated with altered mRNA stability (*Xia et al., 1999*). Vilmundarson et al. demonstrated that HuR differentially modulates *IRF2BP2* translation through a 3′ UTR polymorphism that is associated with increased coronary artery calcification (*Vilmundarson et al., 2021*). Our exploratory studies did not resolve whether the rs13900 allelic variation leads to differences in polysomal loading, and additional studies are required to address this issue. Another important mechanism by which disease-associated genetic variants in the UTRs could influence mRNA stability and translatability is by disrupting or creating microRNA-binding sites (*Huang et al., 2019*). We examined the region flanking the rs13900 and found that there are several predicted binding sites for miRNAs. Among these, several miRNAs have minimum free energy $\leq$ –25 kcal/mol, suggesting that they can potentially play a role in *CCL2* expression and will be investigated in future studies. Of note, a recent study showed that RBPs may play an important role in miRNA-mediated gene regulation (*Kim et al., 2021*).

RBPs regulate the protein expression from a given mRNA by modulation of its half-life, subcellular localization, and ribosomal recruitment (*Glisovic et al., 2008*). However, mRNA abundance and stability are not always predictive of protein synthesis: relative mRNA abundance/stability and translation levels of a given gene can vary significantly and are determined by an assortment of post-transcriptional events (*Liu et al., 2016*). Our previous findings and results from the present study suggest that there is a close association between *CCL2* rs1024611-rs13900 haplotype, mRNA, and protein expression. Nevertheless, we cannot completely rule out the contribution of transcription-based mechanisms to the AEI at the *CCL2* locus. Previous studies have suggested that there is a high level of conservation for interactions between RBPs and their target molecules and that RNA-mediated gene regulation is less evolvable than transcriptional regulation (*Payne et al., 2018*). Thus, it is remarkable that the rs13900 alters HuR binding and impacts both transcript stability and translatability. Our bioinformatic analysis shows that rs13900 could potentially lead to dramatic changes in the secondary structure of *CCL2* mRNA (*Figure 3B*). A recent study using bioinformatic analyses has shown that SNPs may affect RNA-protein interactions from outside binding motifs through altered RNA secondary structure (*Shatoff and Bundschuh, 2020*). We cannot rule out the possibility that HuR may differentially bind to other regions of *CCL2* transcript due to changes in secondary structure and therefore needs to be further examined in future studies.

Given our findings that rs13900 modulates HuR binding and thereby influences *CCL2* expression, targeting HuR-ARE interactions could offer a promising therapeutic avenue for conditions involving heightened monocyte/macrophage recruitment, such as inflammatory, cardiovascular, and neoplastic diseases. Indeed, existing small-molecule inhibitors of HuR (e.g. MS-444, KH-3, and CMLD-2) (*Chaudhary et al., 2023*; *Fattahi et al., 2022*; *de Lange et al., 2017*; *Liu et al., 2020*; *Wang et al., 2019*; *Wei et al., 2024*) highlight the feasibility of disrupting HuR function in vivo, suggesting that co-targeting HuR and the rs13900 may represent a novel precision-based strategy for monocyte/macrophage-driven disorders.

Both rs1024611 and rs13900 are in high LD with rs3091315, which is detected in the GWASs of CD and IBD (*Franke et al., 2010*; *Palmieri et al., 2010*; *de Lange et al., 2017*; *Liu et al., 2015*; *Kenny et al., 2012*). rs3091315 is classified as an upstream risk variant and is located in the intergenic region between *CCL2* and *CCL7*. The GWA risk allele for CD is rs3091315-A, which is correlated with rs1024611(A) and rs13900(C) alleles. While the biological role of *CCL2* in CD and IBD is well recognized (*Maharshak et al., 2010*; *Martin et al., 2019*; *Darkoh et al., 2014*), the results from genetic association studies are not consistent, and the disease outcomes could potentially differ by population studied (*Chen et al., 2016*).

In conclusion, our study shows that the disease associations mediated by the *CCL2* rs1024611-rs13900 haplotype may be due to altered mRNA stability mediated through differential binding of an RBP. Given the importance of mRNA stability in immune homeostasis, such mechanisms could play a critical role in inter-individual differences in disease pathogenesis.

# Materials and methods

## Recruitment of study participants, primary cell culture, and genotyping

All research involving human subjects was approved by the Institutional Review Boards (IRBs) of the following institutions: University of Texas (UT) Health San Antonio, San Antonio, Texas; University of Texas Rio Grande Valley, Edinburg, Texas; and Texas A&M University-San Antonio, San Antonio, Texas. Written informed consent was obtained from each individual for participation in our study in accordance with the IRB of record – UT Health San Antonio IRB (protocol #20160074H). A total of 47 unrelated individuals were recruited into the study. Data and samples from the study participants were obtained at the First Outpatient Research Unit (FORU), UT Health San Antonio.

The rs13900 polymorphism was detected by TaqMan predesigned SNP genotyping assay (Applied Biosystems, CA, USA, Cat. No. 4351379). Briefly, genomic DNA samples were obtained from peripheral blood of healthy study participants using QIAamp DNA Blood Mini Kits (QIAGEN, Cat. No. 51104) in accordance with the manufacturer's protocol, checked for quality and concentration, and stored at −80°C until used. Genotyping of rs13900 was performed on 10 ng of genomic DNA using TaqMan Genotyping Master Mix in a 10 µL reaction volume. PCR was performed on the Quantstudio 12K Flex (Applied Biosystems) in 384-well plates. The temperature cycling conditions consisted of an initial enzyme heat activation step of 10 min at 95°C and 40 cycles of a three-step amplification profile of 20 s at 95°C for denaturation, 1 min at 60°C for annealing, and 30 s at 72°C for extension. QS12K software (Applied Biosystems, CA, USA) was used to score the alleles. Whole blood samples were collected from study participants who were either homozygous (either C/C or T/T) or heterozygous (C/T) for rs13900 at the initial screening for the genotype status, followed by recalling selected individuals for detailed studies. PBMCs were isolated by Ficoll density gradient centrifugation. Total genomic DNA and RNA were isolated from $0.5×10^6$ PBMC following stimulation with 1 µg/mL LPS (Millipore Sigma, Cat. No. L2630) for 3 hr as described previously (*Pham et al., 2012*; *Sharif et al., 2007*). CD14+ monocytes were purified using EasySep Human Monocytes Isolation Kit (negative selection kit; STEMCELL Technologies, Cat. No. 19359) according to the manufacturer's instructions. Monocytes were treated with 50 ng/mL M-CSF (Peprotech, Cat. No. 300-25) for 72 hr to induce macrophage differentiation, and flow cytometric measurement of surface markers CD64 (BD-Pharmigen, Cat. No. 555522), CD206 (BD-Pharmigen, Cat. No. 564060), and CD44 (BD-Pharmigen, Cat. No. 555479) was used to confirm the differentiation.

## RT-qPCR assays

Total RNA was isolated using the RNeasy Plus Mini Kit (QIAGEN, Cat. No. 74134) according to the manufacturer's instructions. 10 µL of total RNA sample (500 ng) was converted to cDNA using MultiScribe Reverse Transcriptase (Applied Biosystems, Cat. No. 4311235) with random primers. Real-time quantitative PCR was performed using TaqMan gene expression assays (Thermo Fisher Scientific, Cat. No. 4331182). Relative expression levels were calculated by applying the $2^{-\Delta\Delta Ct}$ method using 18S rRNA as a reference.

## Capture of nascent RNA

Newly synthesized RNA was isolated using the Click-It Nascent RNA Capture Kit (Invitrogen, Cat. No. C10365) following the manufacturer's protocol. PBMCs or MDMs obtained from heterozygous individuals were stimulated with 1 µg/mL LPS for 3 hr, followed by a 3 hr pulse with 0.2 mM 5-ethynyl uridine (*Jao and Salic, 2008*; *Paulsen et al., 2013*). After the pulse, the culture medium was replaced with fresh growth medium devoid of EU. To assess RNA stability, cells were left untreated or were treated with actinomycin D (5 µg/mL), and samples were collected at 0, 1, 2, and 4 hr post-treatment. The EU RNA was subjected to a click reaction that adds a biotin handle which was then captured by streptavidin beads. The captured RNA was used for cDNA synthesis (Superscript Vilo Kit, Cat. No. 11754250), PCR amplification, and allelic quantification.

## AEI assessment of *CCL2*

Total RNA and genomic DNA were isolated from primary immune cells (PBMC and macrophages) after they were treated with LPS for 3 hr. Total RNA was reverse-transcribed as described above. The region encompassing rs13900 was amplified by PCR in a 50 µL reaction mixture containing 2 µL of cDNA or genomic DNA, 25 µL AmpliTaq Gold 360 Master Mix (Thermo Fisher Scientific, Cat. No. 4398881) and 10 µM each of the forward and reverse primers. The nucleotide sequence of the forward primer and reverse primers was 5′-ACCTGGACAAGCAAACCCAA-3′ and 5′- ACCCTCAAAACATCCC AGGG-3′. The following temperature conditions were used: initial denaturation at 95°C for 10 min followed by 40 cycles of denaturation at 95°C for 30 s, annealing at 53.6°C for 30 s, extension at 72°C for 30 s, and final extension at 72°C for 7 min. The amplicons were purified with the GeneJET PCR Purification Kit (Thermo Fisher Scientific, Cat. No. K0701) and were subjected to Sanger sequencing (Macrogen, USA) using the sequencing primer 5′-GCAAACCCAAACTCCGAAGAC-3′. The degree of AEI for *CCL2* was expressed as a ratio of reference allele (REF; major) to alternative allele (ALT; minor). PeakPicker v.2.0 was used with default settings to quantify the relative amount of the two alleles from the chromatogram after peak intensity normalization (*Ge et al., 2005*).

## Bioinformatic analyses

The ex vivo binding of RBPs to region flanking rs13900 and *CCL2* 3′ UTR was examined using Atlas of UTR Regulatory Activity (AURA) (*Dassi et al., 2014*). The resources included in POSTAR3 were used to interrogate the RNA-binding motifs that overlap the rs13900 (*Zhao et al., 2022*). RBP-var2 was used to explore the potential functional consequences (*Mao et al., 2016*), and the ViennaRNA Package (*Lorenz et al., 2011*) was used to predict changes in *CCL2* secondary structure due to rs13900.

## RNA electrophoretic mobility shift assay

Oligoribonucleotides spanning the rs13900 C or rs13900 T alleles (IDT) labeled with infrared dyes were used to determine the differential binding of HuR. The sequences of the oligoribonucleotide with rs13900 C allele are rCrUrUrUrCrCrCrCrArGrArCrArCrCr**C**rUrGrUrUrUrUrArUrUrU and oligori-bonucleotide with rs13900 T allele is rCrUrUrUrCrCrCrCrArGrArCrArCrCr**U**rUrGrUrUrUrUrArUrUrU. The oligoribonucleotides were incubated with either 10 µg HuR overexpressing HeLa cell whole-cell extracts or 25–200 µM ELAVL1 human recombinant protein (Origene, Cat. No. TP301562) in a 20 µL reaction mixture with 20 units of RNasin and 1× RNA-binding buffer containing 20 mM HEPES (pH 7.6), 3 mM MgCl$_2$, 40 mM KCl, 5% glycerol, 2 mM dithiothreitol, and 20 µg yeast tRNA. The reaction was incubated on ice for 10 min, and super-shift assay was performed by adding 2 µL of anti-HuR antibody (3A2, mouse monoclonal IgG antibody, Santa Cruz Biotechnology, Cat. No. sc-5261). After antibody addition, the complexes were incubated on ice for 15 min and were resolved by electrophoresis

under non-denaturing conditions in 5% polyacrylamide gels with 0.5× TBE running buffer. Gels were then analyzed using Li-Cor Odyssey CLX.

## RNA immunoprecipitation

RNA immunoprecipitation (RIP) was carried out using an immunoprecipitation kit (RNA immunoprecipitation (RIP)-Assay Kit, MBL) following the manufacturer's instructions. Briefly, $2 \times 10^6$ cells were lysed, and the extract was precleared and incubated with 15 µg of antibody against HuR or IgG for 3 hr. RNA on the beads was purified and converted to cDNA using random hexamer. cDNA was further amplified, and the PCR product was sequenced using the following primer: GCAAACCCAAAC TCCGAAGAC.

## Cell culture and transfection

HEK-293 (CRL-1573) cells (gift from Christopher Jenkinson, University of Texas Rio Grande Valley) were maintained at 37°C in 95% air and 5% $CO_2$ and were cultured using Eagle's Minimum Essential Medium (EMEM) supplemented with 10% heat-inactivated FBS, penicillin (100 U/mL), and streptomycin (100 mg, mL) (Gibco). Cultures were routinely screened for mycoplasma contamination. The cells used in the experiments tested negative for mycoplasma. For HuR overexpression and silencing studies, cells were seeded in six-well plates in serum-free EMEM at a density of ~250,000 cells per well. The cells were transfected when they reached ~70–80% confluency, using Lipofectamine 3000 (Thermo Fisher Scientific, Cat. No. L3000008) according to the manufacturer's recommended protocol. Briefly, Lipofectamine 3000 was diluted in OptiMEM and allowed to complex with either 0.5 µg of pCMV6-HuR or pCMV-Entry plasmid or 50 µM of HuR siRNA or control siRNA for 15 min at room temperature. The plasmid/siRNA-Lipofectamine 3000 mixture was then added to the cells in a final volume of 2 mL of complete medium. After incubation for 72 hr, cells were harvested and used for further processing.

## Western blotting

Cell lysates were prepared in chilled RIPA buffer (25 mM Tris-HCl pH 7.6, 150 mM NaCl, 1% NP-40, 1% sodium deoxycholate, 0.1% SDS; Thermo Scientific, Rockford, IL, USA) containing protease inhibitor (Complete Mini, EDTA-free Protease Inhibitor Cocktail Tablets, Roche Diagnostics, Cat. No. 11836170001). Cells were lysed on ice for 10–15 min followed by centrifugation at 12,000×$g$ for 15 min, and the cleared supernatant was collected and stored at −20°C. Cell lysates with equal amounts of total protein (15–20 µg) were loaded and separated on an SDS-polyacrylamide gel (10–15%) and were electrophoretically transferred onto nitrocellulose membranes (Thermo Scientific). The membranes were blocked with non-fat dry milk in TBST (50 mM Tris-HCl, pH 7.4, 150 mM NaCl, 0.2% Tween 20) and were incubated overnight with anti-HuR primary antibodies (diluted 1:1000). Following incubation, the membranes were washed and then further incubated with anti-mouse β-actin for 1 hr at room temperature, followed by HRP-labeled goat anti-mouse or goat anti-rabbit antibodies (1:1000; Santa Cruz). Protein bands were detected using a chemiluminescence (ECL Kit) method (Pierce) and visualized on X-ray film (Kodak).

## Reporter assays

Site-specific mutagenesis was done on *CCL2* LightSwitch 3′ UTR reporter (Active Motif) to generate constructs containing rs13900 C or rs13900 T alleles. The nucleotide sequence of the complete constructs was verified by Sanger sequencing (Macrogen, USA). 0.5 µg of these constructs were nucleofected into HEK-293 cells using 4D Nucleofector (Lonza), and luciferase activity was measured 24 hr post-transfection using SpectraMax plate reader (Molecular Devices). The efficiency of the nucleofection was verified by confocal microscopy, and it was greater than 90%. Given the high efficiency of the nucleofection and to avoid cross-talk with co-transfected control plasmids (*Farr and Roman, 1992*), luciferase data across samples was normalized using the protein content of the lysates using BCA protein assay. mRNA translatability was assessed using methods as described previously (*Zhang et al., 2017*).

## Polysome profiling

Human monocytes were isolated from fresh blood as described earlier (*Gavrilin et al., 2009*) with slight modification. Briefly, PBMCs were isolated by density gradient centrifugation using Histopaque,

followed by immunomagnetic negative selection using EasySep Human Monocyte Isolation Kit. A high purity level for CD14$^+$ cells was consistently achieved (≥90%) through this procedure, as confirmed by flow cytometry. The purified monocytes were immediately used for macrophage differentiation by treating them with 50 ng/mL M-CSF (PeproTech) for 72 hr and flow cytometric measurement of surface markers CD64$^+$, CD206$^+$, and CD44 was used to confirm the differentiation. Cells were treated with LPS or PBS for 3 hr to induce *CCL2* expression. Cells were treated with 0.1 mM cycloheximide, a translation inhibitor, for 3 min, harvested and washed with ice-cold PBS containing 0.1 mM cyclo-heximide, and pelleted by centrifugation for 5 min at 500×*g*. The pellets were either stored at −80°C or immediately used for cytoplasmic lysate preparation. The cell pellet was resuspended in 500 μL lysis buffer containing 150 mM NaCl, 50 mM Tris-HCl pH 7.5, 10 mM KCl, 10 mM MgCl$_2$, 0.2% NP-40, 2 mM dithiothreitol, 2 mM sodium orthovanadate, 1 mM phenylmethylsulfonyl fluoride, and 80 units/mL RNaseOUT. After 10 min incubation on ice with occasional inverting every 2 min, samples were centrifuged at 12,000×*g* for 10 min at 4°C to pellet the nuclei and the post-nuclear supernatants. The optical density at 254 nm was measured, and volumes corresponding to the same OD254 were used. Post-nuclear supernatants were laid on top of a 10–50% sucrose gradient. The gradient was centrifuged for 90 min at 200,000×*g*. After the ultracentrifugation, 10 fractions were collected from top to bottom. RNA was extracted from each fraction and RT-qPCR was performed using *CCL2* mRNA specific primers. For analysis, fraction 1 was considered as cytoplasmic lysate, fractions 2–4 were considered monosomal fraction, and fractions 6–10 as polysomal fraction. The percentage (%) distribution for the *CCL2* mRNAs across the gradients was calculated using the differences in the cycle threshold (ΔCt) values using the following formula (*Nayak et al., 2013*):

% of mRNA A in each fraction = $2^{ΔCt\ fraction\ X} \times 100$/Sum, where ΔCt fraction X=Ct (fraction 1) − CT (fraction X).

## Lentiviral transduction of primary macrophages

Cells were transfected by slightly modifying the method described by *Plaisance-Bonstaff et al., 2019*. Briefly, monocytes were purified from PBMCs obtained from homozygous donors for rs13900 C or rs13900 T by negative selection. Upon purification, cells were resuspended in 24-well plates at a seeding density of 0.5×10$^6$ cells per well and were further cultured in the medium supplemented with 50 ng/mL M-CSF. After 24 hr, ready-to-use GFP-tagged pCMV6-HuR or CMV-null lentiviral particles (Amsbio, Cambridge, MA, USA) were transduced into 0.5×10$^6$ cells in the presence of polybrene (60 μg/mL) at an MOI of 1. The cells were processed for HuR and *CCL2* expression 72 hr after transduction after stimulation with LPS for 3 hr.

## Statistical analyses

All values in figures are presented as the mean ± SEM. Results were analyzed with ANOVA and post hoc contrasts with Fisher's least significant difference (LSD) test or Student's t test. Statistical analyses were carried out in the SigmaPlot 12.0 software.

## Acknowledgements

We thank the participants of the study for their time and support. This research was supported by the National Institute of Allergy and Infectious Diseases (NIAID) of the National Institutes of Health grant 5R01AI119131. The content is solely the responsibility of the authors and does not necessarily represent the official views of NIAID.

## Additional information

### Funding

| Funder | Grant reference number | Author |
| --- | --- | --- |
| National Institute of Allergy and Infectious Diseases | 5R01AI119131 | Srinivas Mummidi |

| Funder | Grant reference number | Author |
|--------|------------------------|--------|

The funders had no role in study design, data collection and interpretation, or the decision to submit the work for publication.

## Author contributions

Feroz Akhtar, Data curation, Formal analysis, Validation, Investigation, Visualization, Methodology, Writing – original draft; Joselin Hernandez Ruiz, Ya-Guang Liu, Liza D Morales, Investigation, Methodology; Roy G Resendez, Data curation, Methodology; Denis Feliers, Alvaro Diaz-Badillo, Investigation, Methodology, Writing – review and editing; Donna M Lehman, Rector Arya, John Blangero, Writing – review and editing; Juan Carlos Lopez Alvarenga, Formal analysis, Methodology, Writing – review and editing; Ravindranath Duggirala, Resources, Formal analysis, Writing – review and editing; Srinivas Mummidi, Conceptualization, Resources, Data curation, Formal analysis, Supervision, Funding acquisition, Visualization, Methodology, Writing – original draft, Project administration, Writing – review and editing

## Author ORCIDs

Feroz Akhtar ⓘ https://orcid.org/0009-0009-1381-3856
Liza D Morales ⓘ https://orcid.org/0000-0003-1056-9121
Srinivas Mummidi ⓘ https://orcid.org/0000-0002-4068-6380

## Ethics

Human subjects: All research involving human subjects was approved by the Institutional Review Boards (IRBs) of the following institutions: University of Texas (UT) Health San Antonio, San Antonio, Texas; University of Texas Rio Grande Valley, Edinburg, Texas; and Texas A&M University- San Antonio, San Antonio, Texas. Written informed consent was obtained from each individual for participation in our study in accordance with the IRB of record-UT Health San Antonio IRB (protocol #20160074H). A total of 47 unrelated individuals were recruited into the study. Data and samples from the study participants were obtained at the First Outpatient Research Unit (FORU), UT Health San Antonio.

Reviewer #1 (Public review): https://doi.org/10.7554/eLife.93108.3.sa1
Reviewer #2 (Public review): https://doi.org/10.7554/eLife.93108.3.sa2
Author response https://doi.org/10.7554/eLife.93108.3.sa3

---

# Additional files

## Supplementary files

Supplementary file 1. Allele carriages and allele frequencies of rs13900 in healthy volunteers. MAF, minor allele frequency.

Supplementary file 2. Crosslinking immunoprecipitation (CLIP) analysis of HuR-binding sites on the 3'untranslated region (3'UTR) of the *CCL2* gene. A summary of the single nucleotide polymorphisms (SNPs) located within HuR-binding regions is shown, including SNP ID (rs#), genomic coordinates, strand, binding score, conservation scores (PhastCons, PhyloP), dataset accession numbers, alleles, and SNP position within peak region. Data were generated using the Genomic Variants Module in the POSTAR3 platform. The GSE accession #s correspond to those reported in the Gene Expression Omnibus (GEO).

Supplementary file 3. Loading of rs13900 alleles to cytosolic, monosomal, and polysomal fractions from macrophage extracts prepared from heterozygous donors. A T:C ratio >1 indicates increased levels of the T allele relative to the C allele.

MDAR checklist

## Data availability

All the data generated for this manuscript are included in this manuscript and supporting files. Source data files have been provided for figures and supplement figures. All reagents used in the experiments are listed in the Key Resources Table in Appendix 1. The plasmids generated as part of this project will be available from the corresponding author upon signing a Material Transfer Agreement.

The following previously published dataset was used:

| Author(s) | Year | Dataset title | Dataset URL | Database and Identifier |
|---|---|---|---|---|
| Lebedeva S, Jens M, Theil K, Schwanhäusser B, Selbach M, Landthaler M, Rajewsky N | 2011 | Unstressed HeLa cells and ELAVL1/HuR knock down conditions: polyA RNA-Seq, small RNA-Seq, and PAR-CLIP | https://www.ncbi.nlm.nih.gov/geo/query/acc.cgi?acc=GSE29943 | NCBI Gene Expression Omnibus, GSE29943 |

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

# Appendix 1

## Appendix 1—key resources table

| Reagent type (species) or resource | Designation | Source or reference | Identifiers | Additional information |
|---|---|---|---|---|
| Cell line (human) | HEK-293 (CRL-1573) | Gift from Christopher Jenkinson, University of Texas Rio Grande Valley | | |
| Transfected construct (human) | GFP-tagged pCMV6-HuR | Amsbio | Cat# LVP071 | |
| Antibody | Anti-human CD14 -Pe-Cy (Mouse monoclonal) | BD Biosciences | Cat# 562698, RRID:AB_2737729 | (5 µL/test) |
| Antibody | Anti-human CD 64-FITC (Mouse monoclonal) | BD Biosciences | Cat# 555527, RRID:AB_395913 | (20 µL/test) |
| Antibody | Anti-human CD206-BV421 (Mouse monoclonal) | BD Biosciences | Cat# 564062, RRID:AB_2738570 | (5 µL/test) |
| Antibody | Anti-human CD11c-AlexaFluor 700 (Mouse monoclonal) | BD Biosciences | Cat# 561352, RRID:AB_10612006 | (5 µL/test) |
| Antibody | Anti-human CD44- PE (Mouse monoclonal) | BD Biosciences | Cat# 555479, RRID:AB_395871 | (20 µL/test) |
| Antibody | Anti-human CD200-APC (Mouse monoclonal) | Thermo Fisher Scientific | Cat# 17-9200-42, RRID:AB_10698161 | (5 µL/test) |
| Antibody | Anti-HuR (3A2) (Mouse monoclonal) | Santa Cruz Biotechnology | Cat# sc-5261, RRID:AB_627770 | REMSA (0.4 µg/rxn) RNA immunoprecipitation (RIP) (15 µg) WB (1:1000) |
| Antibody | β-Actin (2 A3) (Mouse monoclonal) | Santa Cruz Biotechnology | Cat# sc-517582, RRID:AB_2833259 | WB (1:10,000) |
| Antibody | Goat-IgG-control-human | Santa Cruz Biotechnology | Cat# sc-2028, RRID:AB_737167 | RNA immunoprecipitation (RIP) (15 µg) |
| Antibody | Goat anti-mouse IgG-HRP (Goat polyclonal) | Santa Cruz Biotechnology | Cat# sc- 2005, RRID:AB_3717730 | WB (1:1000) |
| Antibody | Goat anti-rabbit IgG-HRP (Goat polyclonal) | Santa Cruz Biotechnology | Cat# sc-2004, RRID:AB_631746 | WB (1:1000) |
| Recombinant DNA reagent | MultiScribe Reverse Transcriptase | Thermo Fisher Scientific | 4311235 | |
| Sequence-based reagent | rs13900_F | ACCTGGACAAGCAAACCCAA | PCR primers | |
| Sequence-based reagent | rs13900_R | ACCCTCAAAACATCCCAGGG | PCR primers | |
| Sequence-based reagent | rs13900 | GCAAACCCAAACTCCGAAGAC | Sequencing primer | |
| Sequence-based reagent | rs13900C | rCrUrUrUrCrCrCrArGrArCrArC rCrCrUrGrUrUrUrArUrU | Oligoribonucleotide | |
| Sequence-based reagent | rs13900C | rCrUrUrUrCrCrCrArGrArCrArC rCrUrGrUrUrUrUrArUrU | Oligoribonucleotide | |
| Peptide, recombinant protein | Recombinant Human M-CSF | Peprotech | Cat. #: 300-25 | 50 ng/mL |
| Peptide, recombinant protein | ELAVL1 human recombinant protein | OriGene | Cat. #: TP301562 | |
| Commercial assay or kit | EasySep Human Monocyte Isolation Kit | STEMCELL Technologies | STEMCELL Technologies: 19359 | |
| Commercial assay or kit | QIAamp DNA Blood Mini Kit (50) | QIAGEN | QIAGEN: 51104 | |

*Appendix 1 Continued on next page*

*Appendix 1 Continued*

| Reagent type (species) or resource | Designation | Source or reference | Identifiers | Additional information |
|---|---|---|---|---|
| Commercial assay or kit | RNeasy Plus Mini Kit (50) | QIAGEN | QIAGEN: 74134 | |
| Commercial assay or kit | Click-iT-Nascent RNA Kit | Thermo Fisher Scientific | Thermo Fisher Scientific: C10365 | |
| Commercial assay or kit | Superscript Vilo cDNA Synthesis Kit | Thermo Fisher Scientific | Thermo Fisher Scientific: 11754250 | |
| Commercial assay or kit | GeneJET PCR Purification Kit | Thermo Fisher Scientific | Thermo Fisher Scientific: 1 K0701 | |
| Commercial assay or kit | RIP-Assay Kit | MLB Life Sciences | MLB Life Sciences: RN1001 | |
| Commercial assay or kit | LightSwitch Lucifgerase Assay Kit | Active Motif | Active Motif: 32032 | |
| Commercial assay or kit | TaqMan SNP Genotyping Assay | Thermo Fisher Scientific | Thermo Fisher Scientific: 4351379 | Assay id: C___7449810_10 |
| Commercial assay or kit | TaqMan Gene Expression Assay (FAM) | Thermo Fisher Scientific | Thermo Fisher Scientific: 4331182 | Assay id: Hs03003631_g1 |
| Commercial assay or kit | TaqMan Gene Expression Assay (FAM) | Thermo Fisher Scientific | Thermo Fisher Scientific: 4331182 | Assay id: Hs00736046_m1 |
| Commercial assay or kit | TaqMan Genotyping Master Mix | Thermo Fisher Scientific | Thermo Fisher Scientific: 43711355 | |
| Commercial assay or kit | TaqMan Fast Advanced Master Mix | Thermo Fisher Scientific | Thermo Fisher Scientific: 4444964 | |
| Commercial assay or kit | AmpliTaq Gold 360 Master Mix | Thermo Fisher Scientific | Thermo Fisher Scientific: 4398881 | |
| Chemical compound, drug | Histopaque | Millipore Sigma | 10771 | |
| Chemical compound, drug | Lipopolysaccharides from *Escherichia coli* | Millipore Sigma | L2630 | 1 µg/mL |
| Chemical compound, drug | Actinomycin D | Millipore Sigma | A1410 | 5 µg/mL |
| Chemical compound, drug | Lipofectamine 3000 | Thermo Fisher Scientific | L3000008 | |
| Chemical compound, drug | cOmplete, Mini, EDTA-free Protease Inhibitor Cocktail | Millipore Sigma | 118361700 | |
| Chemical compound, drug | $MgCl_2$ (magnesium chloride) (25 mM) | Thermo Fisher Scientific | R0971 | |
| Chemical compound, drug | KCl (2 M), RNase-free | Thermo Fisher Scientific | AM9640G | |
| Chemical compound, drug | Glycerol, Molecular Biology Grade | Thermo Fisher Scientific | J61059AP | |
| Chemical compound, drug | Dithiothreitol (DTT) | Thermo Fisher Scientific | D1532 | |
| Chemical compound, drug | Penicillin-Streptomycin (10,000 U/mL) | Thermo Fisher Scientific | 15140122 | 100 U/mL |
| Chemical compound, drug | Pierce ECL Western Blotting Substrate | Thermo Fisher Scientific | 32106 | |
| Chemical compound, drug | Pierce Protein G Agarose | Thermo Fisher Scientific | 20398 | |
| Software, algorithm | ImageJ | ImageJ | RRID:SCR_003070 | |
| Software, algorithm | SigmaPlot | SigmaPlot | RRID:SCR_003210 | |
| Software, algorithm | QS12K Real-Time PCR Software | Thermo Fisher Scientific | | |

*Appendix 1 Continued on next page*

*Appendix 1 Continued*

| Reagent type (species) or resource | Designation | Source or reference | Identifiers | Additional information |
|---|---|---|---|---|
| Software, algorithm | PeakPicker v.2.0 | PeakPicker | https://doi.org/10.1101/gr.4023805 | |
| Software, algorithm | AURA | Atlas of UTR Regulatory Activity | https://doi.org/10.1093/bioinformatics/btr608 | |
| Software, algorithm | POSTAR3 | POSTAR3 | https://doi.org/10.1093/nar/gkab702 | |
| Software, algorithm | RBP-var2 | RBP-var2 | https://doi.org/10.1093/nar/gkv1308 | |
| Software, algorithm | ViennaRNA Package | ViennaRNA | RRID:SCR_008550 | |

