## [Editor Report · eLife Assessment]

CCL2 is a chemokine with immune cell chemoattractant properties, and it appears to play a role in several chronic inflammatory diseases. The RNA-binding protein HuR controls the stability and translation of CCL2 mRNA. This paper presents **convincing** evidence that a relatively common genetic variant tied to several disease phenotypes affects the interaction between the mRNA of CCL2 and the RNA-binding protein HuR. While the experiments cannot definitively distinguish between effects on RNA transcription and stability, CCL2 is thought to be relevant for leukocyte migration in various conditions, including chronic inflammation and cancer, and the study presents **important** findings that may be relevant to a broad audience.

---

## [Referee Report · Reviewer #1 (Public review)]

Summary:

This paper presents evidence that a relatively common genetic variant tied to several disease phenotypes affects the interaction between the mRNA of CCL2 and the RNA binding protein HuR. CCL2 is an immune cell chemoattractant protein.

Strengths:

The study is well conducted with relevant controls. The techniques are appropriate, and several approaches provided concordant results were generally supportive of the conclusions reached. The impact of this work, identifying a genetic variant that works by altering the binding of an RNA-regulatory protein, has important implications given that the HuR protein could be a drug target to improve its function and over-ride this genetic change. This could have important implications for a number of diseases where this genetic variant contributes to disease risk.

The authors have done a nice job of citing prior work. Details of the experimental protocols are well elaborated and the significance of the findings are well contextualized.

Weaknesses:

Authors have addressed prior weaknesses.

---

## [Referee Report · Reviewer #2 (Public review)]

This study focuses on the differential binding of the RNA-binding protein HuR to CCL2 transcript (genetic variants rs13900 T or C). The study explores how this interaction influences the stability and translation of CCL2 mRNA. Employing a combination of bioinformatics, reporter assays, binding assays, and modulation of HuR expression, the study proposes that the rs13900T allele confers increased binding to HuR, leading to greater mRNA stability and higher translational efficiency. These findings indicate that rs13900T allele might contribute to heightened disease susceptibility due to enhanced CCL2 expression mediated by HuR. The study is interesting and most results are convincing, however the interpretation relative to RNA transcription and/or stability must be modified, and some data need better presentation or interpretation.

Major Points

Figure 2C:

The authors describe an experiment to assess mRNA stability by labeling nascent RNA with EU for 3 hours, followed by washout of EU, and then incubation with or without actinomycin D for an additional 4 hours before measuring the remaining EU-labeled RNA. While the approach to label nascent RNA with EU is appropriate for tracking RNA decay, I have concerns regarding the use and interpretation of actinomycin D in this context.

After EU washout, the pool of EU-labeled RNA is fixed and no new EU incorporation can occur. Therefore, the addition of actinomycin D at this stage should not affect the decay rate of the already labeled RNA, as transcription of EU-labeled RNA has effectively ceased. In this design, measuring the decrease in EU-labeled RNA over time reflects mRNA stability (even in absence of actinomycin D) rather than transcriptional activity.

Therefore, the authors' statement that the non-actinomycin D treatment group represents transcriptional changes is not accurate here. Since EU labeling was stopped prior to the 4-hour incubation, any changes in EU-labeled RNA levels during this period reflect RNA decay, not new transcription.

In summary:

To assess transcriptional changes, one would compare the amount of EU-labeled RNA synthesized during the initial labeling period (the first 3 hours), before washout.

If the authors wish to use actinomycin D to block transcription, this should be done in a separate decay assay without EU labeling.

In the current experimental setup, actinomycin D is unnecessary after EU washout and does not influence the decay of the labeled RNA.

I recommend the authors reconsider the interpretation of their data accordingly. I recommend to remove the data points relative to the presence of actinomycin D, as the non-actinomycin D samples are already representative of post-transcriptional changes given that EU was washed out. If Authors want to assess transcriptional changes, they would have to assess the levels during the initial labeling period (before the washout). Transcriptional differences were not assessed, therefore I would modify the text accordingly.

In this context, any changes observed in the actinomycin D-treated samples are likely attributable to general cellular stress induced by actinomycin D, which is known to be highly stressful for cells. This stress could indirectly influence the decay rates of already-labeled EU-RNA.

Figure 4C and 4D:

The Author provided an updated gel with relative quantification - which effectively show the enhanced binding of CCL2 mRNA carrying the T variant to HuR - but they only provided it as data for reviewers (Figure R1). I highly recommend to use these data in the final manuscript instead of the data currently presented in Figure 4C and 4D. This would be important in order not to not create confusion in the reader or concerns regarding probe degradation or saturation.

Minor points

For the IP, I recommend to explain in the final version why the input was not provided (lack of material) and to clarify that the specific binding of Actin was used as a loading control in absence of input. This would be highly beneficial for the readers.

---

## [Author Response]

The following is the authors’ response to the original reviews

**Reviewer #1 (Recommendations For The Authors):**
Comment 1: The authors need to do more to cite the prior work of others. CCL2 allelic expression imbalance tied to the rs13900 alleles was first reported by Johnson et al. (Pharmacogenet Genomics. 2008 Sep; 18(9): 781-791) and should be cited in the Introduction on line 128 next to the Pham 2012 reference. Also, in the Results section, line 142, please provide references for the statement "We and others have previously reported a perfect linkage disequilibrium between rs1024611 in the CCL2 cis-regulatory region and rs13900 in its 3′ UTR" since the linkage disequilibrium for these 2 SNPs is not reported in the ENSEMBL server for the 1000 genomes dataset. #

We thank the reviewer for pointing out the omission regarding the citation of prior work. We acknowledge that Johnson et al. (2008) reported the association between rs13900 and *CCL2* allelic expression imbalance based on Snapshot methodology while examining _cis-_acting variants of 42 candidate genes. To acknowledge these prior studies, we have cited the previous works of Johnson et al. (Johnson et al., 2008) along with Pham et al. (Pham et al., 2012) that linked rs13900 to CCL2 allelic expression imbalance. The text in the introduction section (Lines 128-130) has been updated to reflect the above-mentioned changes.

“We and others have demonstrated AEI in CCL2 using rs13900 as a marker with the T allele showing a higher expression level relative to C allele (Johnson et al., 2008; Pham et al., 2012).”

We have cited some previous studies that suggested strong linkage disequilibrium between rs1024611 and rs13900 within *CCL2* gene, with D′=1 and R^2^=0.96 (Hubal et al., 2010; Intemann et al., 2011; Kasztelewicz et al., 2017; Pham et al., 2012) on Line 144. To address the concern regarding unreported linkage disequilibrium between rs1024611 and rs13900, we reviewed the pairwise linkage disequilibrium data by population in the ENSEMBL server for 1000 Genome dataset and confirm that the linkage disequilibrium (LD) between rs1024611 and rs13900 has been observed, with D′=1 and R^2^=0.92 to 1.0 in specific populations. We have included a table (Author response table 1) depicting pairwise LD between rs13900 and rs1024611 as reported in the ENSEMBL server for the 1000 genome dataset, a URL reference to the ENSEMBL server data.

**Author response table 1. sa3table1:** Pairwise linkage disequilibrium data between rs13900 and rs1024611 by population reported in the ENSEMBL server for the 1000 genome dataset.

Population	Description	Focus Variant	Variant 2	r2	D^(')
1000GENOMES:phase_3:PEL	Peruvian in Lima, Peru	rs1024611	rs13900	0.92478	1
1000GENOMES:phase_3:YRI	Yoruba in Ibadan, Nigeria	rs1024611	rs13900	0.97254	1
1000GENOMES:phase_3:GWD	Gambian in Western Division, The... (more)	rs1024611	rs13900	0.9737	1
1000GENOMES:phase_3:FIN	Finnish in Finland	rs1024611	rs13900	0.97835	1
1000GENOMES:phase_3:JPT	Japanese in Tokyo, Japan	rs1024611	rs13900	0.97905	1
1000GENOMES:phase_3:CHB	Han Chinese in Bejing, China	rs1024611	rs13900	0.97919	1
1000GENOMES:phase_3:ACB	African Caribbean in Barbados	rs1024611	rs13900	1	1
1000GENOMES:phase_3:ASW	African Ancestry in Southwest US	rs1024611	rs13900	1	1
1000GENOMES:phase_3:BEB	Bengali in Bangladesh	rs1024611	rs13900	1	1
1000GENOMES:phase_3:CDX	Chinese Dai in Xishuangbanna, China	rs1024611	rs13900	1	1
1000GENOMES:phase_3:CEU	Utah residents with Northern and... (more)	rs1024611	rs13900	1	1
1000GENOMES:phase_3:CHS	Southern Han Chinese, China	rs1024611	rs13900	1	1
1000GENOMES:phase_3:CLM	Colombian in Medellin, Colombia	rs1024611	rs13900	1	1
1000GENOMES:phase_3:ESN	Esan in Nigeria	rs1024611	rs13900	1	1
1000GENOMES:phase_3:GBR	British in England and Scotland	rs1024611	rs13900	1	1
1000GENOMES:phase_3:GIH	Gujarati Indian in Houston, TX	rs1024611	rs13900	1	1
1000GENOMES:phase_3:IBS	Iberian populations in Spain	rs1024611	rs13900	1	1
1000GENOMES:phase_3:ITU	Indian Telugu in the UK	rs1024611	rs13900	1	1
1000GENOMES:phase_3:KHV	Kinh in Ho Chi Minh City, Vietnam	rs1024611	rs13900	1	1
1000GENOMES:phase_3:LWK	Luhya in Webuye, Kenya	rs1024611	rs13900	1	1
1000GENOMES:phase_3:MSL	Mende in Sierra Leone	rs1024611	rs13900	1	1
1000GENOMES:phase_3:MXL	Mexican Ancestry in Los Angeles... (more)	rs1024611	rs13900	1	1
1000GENOMES:phase_3:PJL	Punjabi in Lahore, Pakistan	rs1024611	rs13900	1	1
1000GENOMES:phase_3:PUR	Puerto Rican in Puerto Rico	rs1024611	rs13900	1	1
1000GENOMES:phase_3:STU	Sri Lankan Tamil in the UK	rs1024611	rs13900	1	1

F. Variant, Focus Variant; R^2^, correlation between the pair loci; D′, difference between the observed and expected frequency of a given haplotype.

URL: https://www.ensembl.org/Homo_sapiens/Variation/HighLD?db=core;r=17:34252269-34253269;v=rs1024611;vdb=variation;vf=959559590;second_variant_name=rs13900

Comment 2: Certain details of the experimental protocols need to be further elaborated or clarified to contextualize the significance of the findings. For example, in the results line 184 the authors state "Using nascent RNA allows accurate determination of mRNA decay by eliminating the effects of preexisting mRNA." How does measuring nascent RNA enable the accurate determination of mRNA decay? Doesn't it measure allele-specific mRNA synthesis? Please elaborate, as this is a key result of the study. Can the authors provide a reference supporting this statement?

It is worthwhile to mention that mRNA decay can be precisely measured by eliminating the effect of any preexisting mRNA. Metabolic labeling with 4-thiouridine allows exclusive capture of newly synthesized RNA which will allow quantification of RNA decay eliminating any interference from preexisting RNA. We agree that nascent RNA measurement primarily reflects synthesis rate rather than degradation. However, in conjugation with actinomycin-D based inhibition studies it can be exploited for accurate mRNA decay determination of the newly synthesized RNA (Russo et al., 2017). Therefore, our aim was to use the nascent RNA to study decay kinetics. The imbalance in the *CCL2* allele expression does occur at the transcriptional level as seen in non-actinomycin-D treatment group (Figure 2C) although the impact of post-transcriptional mechanisms that alter transcripts stability cannot be ruled out. Therefore, we employed a novel approach that could assess both the synthesis and the degradation by combining actinomycin-D inhibition and nascent RNA capture in the same experimental setup. In the presence of actinomycin-D, we could detect much greater allelic difference in the expression levels of the rs13900T and C allele four-hour post-treatment, suggesting a role for post-transcriptional mechanisms in CCL2 AEI.

“We have expanded the method section in the revised draft to include experimental details on capture of nascent RNA and subsequent downstream analysis” (Lines 553-563).

Newly synthesized RNA was isolated using the Click-It Nascent RNA Capture Kit (Invitrogen, Cat No: C10365) following the manufacturer’s protocol. Peripheral blood mononuclear cells (PBMCs) or monocyte-derived macrophages (MDMs) obtained from heterozygous individuals were stimulated with lipopolysaccharide (LPS) for 3 hours in presence of 0.2 mM 5-ethynyl uridine (EU) (Jao and Salic, 2008; Paulsen et al., 2013). After the pulse, the culture medium was replaced with fresh growth medium devoid of EU. To assess RNA stability, actinomycin-D (5 µg/mL) was added, and samples were collected at 0, 1, 2, and 4 h post-treatment. The EU RNA was subjected to a click reaction that adds a biotin handle which was then captured by streptavidin beads. The captured RNA was used for cDNA synthesis (Superscript Vilo kit, Cat No: 11754250), PCR amplification, and allelic quantification.”

Comment 3: Also, they next state that the assay was carried out using cells treated with actinomycin D (line 186). Doesn't actinomycin D block transcription? The original study by Jia et al 2008 in PNAS reported that low concentration of ActD (100 nM) blocked RNA pol I and higher concentration (2 uM) blocked RNA pol II. This or the study on which the InVitrogen kit is based should be cited. The concentration of actinomycin D used to treat the cells should be given. They report that the T allele transcript was more abundant than the C allele transcript in nascent RNA. Why doesn't that argue for a transcriptional mechanism rather than an RNA-stability mechanism? This result should be discussed in the Discussion.

In our study, we used a concentration of 5 µg/mL (3.98 µM), which as noted by the reviewer can effectively inhibit RNA polymerase II (Pl II) activity. We have updated our manuscript to include details and cited the original work of (Jao and Salic, 2008; Paulsen et al., 2013), which thoroughly investigate the effect of various concentrations of ActD on RNA polymerase I and II (Line no 557). A discussion of the RNA stability mechanism is provided in the Result section (Lines 196-198).

Comment 4: In their bioinformatics analysis of the allele-specific CCL2 mRNAs, they reported that the analysis obtained a score of 1e (line 214). What does that mean? Is it significant?

We acknowledge that the notation “a score of 1e” was unclear and thank the reviewer for pointing it out. We have clarified its significance in the revised manuscript. The following text has been included in the result section (Line no 223)

“The score of 1e was obtained using RBP-Var, a bioinformatics tool that scores variants involved in posttranscriptional interaction and regulation (Mao et al., 2016). Here, the annotation system rates the functional confidence of variants from category 1 to 6. While Category 1 is the most significant category and includes variants that are known to be expression quantitative trait loci (eQTLs), likely affecting RBP binding site, RNA secondary structure and expression, category 6 is assigned to minimal possibility to affect RBP binding. Additionally, subcategories provide further annotation ranging from the most informational variants (a) to the least informational variant (e). Reported 1e denotes that the variant has a motif for RBP binding. Although the employed scoring system is hierarchical from 1a to 1e, with decreasing confidence in the variant’s function. However, all the variants in category 1 are considered potentially functional to some degree.”

Comment 5: In Figure 3A, why is the rare SNP rs181021073 shown? This SNP does not comeup anywhere else in the paper. For clarity, it should be removed from Figure 3A.

We thank the reviewer for pointing out the error in Figure 3A and apologize for the oversight. We agree that the SNP rs1810210732 is not mentioned anywhere in the manuscript and its inclusion in Figure 3A may have caused confusion. We have removed this SNP from the revised figure.

Comment 6: For the RNA EMSA results presented in Fig. 4C with recombinant ELAVL1 (HuR), there is clearly a loss of unbound T allele probe with increasing concentrations of the recombinant protein (without a concomitant increase in shifted complex). This suggests that the T allele probe is degraded or loses its fluorescent tag in the presence of recombinant HuR, whereas the C allele probe does not. The quantitation of the shifted complex presented in Fig. 4D as a percentage of bound and unbound probe is therefore artificially elevated for the T allele compared to the C allele. In fact, there seems to be little difference between the shifted complexes with the T and C allele probes. The authors should explain this difference in free probe levels.

We appreciate the constructive critique of the reviewer regarding the RNA EMSA results in Fig. 4C. To address this, we repeated the experiments to analyze the differential binding of rs13900T/C allele bearing probes with increasing concentration of the recombinant HuR. No degradation/ loss of fluorescence tag for T allele was noted in presence of recombinant HuR in three independent experiments (Author response image 1). This indicates that both the probes with C or T allele show comparable stability and are not affected by increasing concentration of recombinant HuR. The apparent reduction in the unbound T allele probe in Figure 4C may be due to saturation at higher HuR concentration rather than degradation.

**Author response image 1. sa3fig1:** Differential binding and stability of oligoribonucleotide probes containing rs13900C or T alleles with recombinant HuR. (A) REMSA with labeled oligoribonucleotides containing either rs13900C or rs13900T and recombinant HuR at indicated concentrations. (B&C) Representative quantitative densitometric analysis of HuR binding to the oligoribonucleotides bearing rs13900 T or C. The signal in the bound fractions were normalized with the free probe. The figure represents data from three independent experiments (mean ± SEM).

Comment 7: In the Methods section, concentrations and source of reagents should be given. For example, what was the bacterial origin of LPS and concentration? What concentration of actinomycin D? What was the source? Was it provided with the nascent RNA kit? In describing the riboprobes used for REMSA, please underline the allele in the sequences (lines 549 and 550).

Thank you for your detailed feedback and suggestions regarding the Materials and Methods Section. We regret the oversight in providing detailed information on reagent concentrations and sources in the method section. We have now rectified this omission and have provided the necessary details and a summary of material/reagents used is presented as a supplementary table (Supplementary Table 4) to enable others to replicate our experiments accurately. Regarding the description of riboprobes for RNA Electrophoretic Mobility Shift Assay, we underlined and bold the allele in the sequences as suggested (Lines 603-604).

Comment 8: For polysome profiling on line 603, please provide a protocol for the differentiation of primary macrophages from monocytes (please cite an original protocol, not a prior paper that does not give a detailed protocol).

We agree with the reviewer’s comment and have included the following text for primary macrophage differentiation from monocytes in the method section cited the original protocol (Line 668).

“Human monocytes were isolated from fresh blood as described earlier (Gavrilin et al., 2009) with slight modification. Briefly, peripheral blood mononuclear cells were isolated by density gradient centrifugation using Histopaque, followed by immunomagnetic negative selection using EasySep Human Monocyte isolation kit. A high purity level for CD14+ cells was consistently achieved (≥90%) through this procedure, as confirmed by flowcytometry. The purified monocytes were immediately used for macrophage differentiation by treating them with 50 ng/mL M-CSF (PeproTech) for 72 h and flow cytometric measurement of surface markers CD64+,

CD206+, CD44 was used to confirm the differentiation”. This data is now shown in the new Supplementary Figure 5—figure supplement 1 and Figure 8—figure supplement 2.

Comment 9: In the legend of Figure 2, please replace "5 ug of actinomycin D" with the actual concentration used.

We appreciate your attention to detail and thank you for pointing out the error in the legend of Figure 2. We regret the oversight and have made the suggested change (Line 739).

Comment 10: In the Discussion, the authors cite the study of CCL2 mRNA stabilization by HuR in mice by Sasaki et al (lines 407-9). Is regulation of CCL2 mRNA by HuR in the mouse relevant to human studies?How conserved is the 3′UTR of mouse and human CCL2? Is the rs13900 variant located in a conserved region? How many putative HuR sites are found in the 3′UTR of human and mouse CCL2 3′UTR? Does HuR dimerize (see Pabis et al 2019, NAR)? This information could be added to the Discussion.

Thank you for your valuable comment. We appreciate your suggestion to include information on the dimerization of HuR in our discussion. While reporting the overall structure and domain arrangement of HuR, Pabis et al. (2019) deciphered dimerization involving Trp261 in RRM3 as key requirement for functional activity of HuR in vitro. This finding provides additional context for understanding HuR’s role in regulating *CCL2* expression. We have added the following few lines in the discussion (Lines 421-428) acknowledging HuR’s ability to dimerize and cite the relevant references.

“HuR consists of three RNA recognition motifs (RRMs) that are highly conserved and canonical in nature (Ripin et al., 2019). In absence of RNA the three RRMs are flexibly linked but upon RNA binding they transition to a more compact arrangement. Mutational analysis revealed that HuR function is inseparably linked to RRM3 dimerization and RNA binding. Dimerization enables recognition of tandem AREs by dimeric HuR (Pabis et al., 2019) and explains how this protein family can regulate numerous targets found in pre-mRNAs, mature mRNAs, miRNAs and long noncoding RNAs.”

We aligned the *CCL2* 3′UTR from five different mammalian species and found that the region flanking rs13900/ HuR binding site is relatively conserved (Author response image 2). Based on PAR-CLIP datasets there are four HuR binding regions in human *CCL2* 3′ UTR (Lebedeva et al., 2011). However, the region overlapping rs13900 seems to be predominantly involved in the *CCL2* regulation (Fan et al., 2011). This information has been included in the discussion.

**Author response image 2. sa3fig2:** Cross-species alignment of the *CCL2* 3′ UTR region flanking the rs13900 using homologous regions from 5 different mammals. (Hu, Human; CH, Chimps; MO, Mouse; RA, Rat; DO, Dog, rs13900 is shown within the brackets Y, pyrimidine)

**Reviewer #2 (Recommendations For The Authors):**
Comment 1: The supplemental figures need appropriate figure legends.

We regret the oversight and thank the reviewer for bringing it to our attention. We have now included the figure legend for the supplemental figures in the revised manuscript.

Comment 2: The data on LPS-induced CCL2 expression in PBMCs should be represented as a scatter plot with statistical significance to enhance clarity and interpretability.

We thank the reviewer for this constructive suggestion. In the revised Figure 2A the induction of *CCL2* expression by LPS in PBMCs obtained from 6 volunteers is represented as a scatter plot. We have also included individual data points in the updated figure and statistical significance to improve clarity and interpretability.

Comment 3: The stability of CCL2 mRNA in control cells needs comparison with treated cells for context. The stability of a housekeeping gene (such as GAPDH or ACTB) should always be included as a control in actinomycin D experiments. Clarify the differential stability of rs13900C vs. rs13900T alleles.

We used 18S to normalize data for the mRNA stability studies, as it is abundant and has been recommended for such studies, as it is relatively unaltered when compared to other housekeeping genes following Act D treatment in well-controlled studies (Barta et al., 2023). We also compared Ct values between the Act D-treated samples and the Act D-untreated samples in this study and found them to be comparable (Author response image 3).

**Author response image 3. sa3fig3:** Ct values of 18s rRNA in ACT-D and control samples in Fig 2.

Comment 4: In the main text and the methods, the authors state that nascent RNA was obtained in the presence of actinomycin D and EU. However, actinomycin D blocks the transcription of nascent RNAs, therefore the findings in Figure 2C do not reflect nascent RNA

Please see our response to Reviewer 1 Comment 2. We would like to emphasize that to assess the differential role of the rs13900 in nascent RNA decay we integrated nascent RNA labeling and transcriptional inhibition. Briefly, PBMC from a heterozygous individual were either unstimulated or stimulated with LPS and pulsed with *5-ethynyl uridine* (0.2 mM) for 3 h and the media was replaced with EU free growth medium. RNA was obtained at 0,1, 2 and 4 h following actinomycin-D treatment (5 µg/mL) to assess the stability of nascent RNA.

Comment 5: Figure 4A is not clearly described or labeled. What are lanes 2 and 6?

Figure 4 has now been updated to clearly describe all the lanes. Lanes 2 and 6 represent the mobility shift seen following the incubation by whole cell extracts and oligonucleotide bearing rs13900C and rs13900T probes respectively.

Comment 6: Figure 4C and Figure 4D: the charts in Figure 4D do not seem to reflect the changes in Figure 4C. How was the mean variant calculated? How do the authors explain the different quantities in unbound/free RNA in rs13900C compared to rs13900T?

We appreciate the constructive critique of the reviewer regarding the RNA EMSA results in Fig. 4C. To address this, we repeated the experiments to analyze the differential binding of rs13900T/C probes with increasing concentration of the recombinant HuR. No degradation/ loss of fluorescence tag in presence of HuR was noted in case of T allele (Author response image 1). This indicates that both the C and T allele probes exhibit comparable stability and are not affected by increasing the concentration of recombinant HuR. The apparent reduction in the unbound T allele probe in Figure 4C may be due to saturation due to higher HuR concentration rather than degradation. Also please note under limiting HuR concentration (50µM) there is more binding of purified HuR by the T bearing oligoribonucleotide (compare lanes 2 & 6 in Author response image 1).

Comment 7: Figure 5A does not look like an IP. The authors should show the heavy and light chains and clarify why there is co-precipitation of beta-actin with IgG and HuR. Also, they should include input samples. Figure 5B: given that in a traditional RIP the mRNA is not cross-linked and fragmented, any region of CCL2 mRNA would be amplified, not just the 3′UTR. In other words, Figure 5B can be valuable to show the enrichment of CCL2 mRNA in general, but not the enrichment of a specific region.

We understand the reviewer’s concern on Figure 5A and 5B. Due to sample limitations we are unable to confirm these results using heavy and light chains antibodies. However, it is important to note that co-precipitation of β-actin with IgG and HuR can be due to its non-specific binding with protein G. In a recent study non-specific precipitation by protein G or A was reported for proteins such as p53, p65 and β-actin (Zeng et al., 2022). We are including a figure provided by MBL Life Sciences as the quality check document for their RIP Assay Kit (RN 1001) that was used in our study. It is evident from Author response image 4 that even pre-clearing the lysate may not remove the ubiquitously expressed proteins such as β-actin or GAPDH and they will persist as contaminants in pull-down samples. Hence the presence of β-actin in the IgG and HuR IP fractions may be due to non-specific interactions with the agarose beads.

**Author response image 4. sa3fig4:** MBL RIP-Assay Kit’s Quality Check. Quality check of immunoprecipitated endogenous PTBP1 expressed in Jurkat cells. Lane 1: Jurkat (WB positive cells), Lane 2: Jurkat + normal Rabbit IgG, Lane 3: Jurkat+ anti-PTBP1.

We agree with the reviewer’s comments that traditional RIP without cross-linking and fragmentation allows amplification of any region of *CCL2* mRNA. However, the upregulation of *CCL2* gene expression in α-HuR immunoprecipitated samples indirectly reflects the enrichment of *CCL2* mRNA associated with HuR. Moreover, 3′-UTR targeting primers were used for amplification to examine HuR binding at this region. We believe this approach ensures that the above enrichment specifically reflects HuR association with the 3′-UTR rather than other parts of the transcript.

Comment 8: Construct Validation in Luciferase Assays (Figure 6): The authors need to confirm equal transfection amounts of constructs and show changes in luciferase mRNA levels. It would be better to use a dual luciferase construct for internal normalization.

We would like to thank the reviewer for his concern and comments related to the luciferase reporter assay. As mentioned in the Methods equal transfection amount (0.5 µg) were used in our study (Line 658). We chose to normalize the reporter activity using total protein concentration instead of using a dual-reporter system to avoid crosstalk with co-transfected control plasmids. This is now included in the Materials and Method section (Lines 662-664). The optimized design of the LightSwitch Assay system used in our study allows a single assay design when a highly efficient transfection system is used (as recommended by the manufacturer). We verified the presence of the correct insert in the *CCL2* Light Switch 3′UTR reporter constructs (Author response image 5). We also sequenced the vector backbone of both constructs to rule out any inadvertently added mutations.

**Author response image 5. sa3fig5:** Schematic of the Lightswitch 3′ UTR vector. (A) Vector information. The vector contains a multiple cloning site (MCS) upstream of the Renilla Luciferase gene (RenSP). Human 3′ UTR *CCL2* is cloned into MCS downstream of the reporter gene and it becomes a part of a hybrid transcript that contains the luciferase coding sequence used to the UTR sequence of *CCL2*. Constructs containing rs13900C or rs13900T allele were generated using site-specific mutagenesis on *CCL2* LightSwitch 3′ UTR reporter. The constructs were validated by Sanger sequencing. (B&C) Sequence chromatograph of the constructs containing *CCL2*-3′UTR insert showing rs13900C and rs13900T respectively. The result confirms the fidelity of the constructs used in the reporter assay.

Comment 9: Polysome Data Presentation: The authors should present the distribution of luciferase mRNA (rs13900T and rs13900C) in all fractions separately and include data on the translation of a control like ACTB or GAPDH.

Since our assessment of *CCL2* allele-specific enrichment in the polysome fractions from MDMs of heterozygous donors did not yield a consistent pattern for differential loading (Supplementary Table3), we used a 3′UTR reporter-based assays that estimated the impact of rs13900 T and C alleles on overall translational output (translatability). The translatability was calculated as luciferase activity normalized by luciferase mRNA levels after adjusting for protein and 18S rRNA using a previously reported method (Zhang et al., 2017). As the measurement of relative allele enrichment in polysome fractions was not included in our invitro reporter assays, it is not possible to present the distribution of luciferase mRNA in various fractions separately. Author response image 6 shows the proportion of *CCL2* mRNA in different fractions corresponding to cytosolic, monosome and polysome fractions obtained from MDM lysates from heterozygous donors along with 18S rRNA quantification.

**Author response image 6. sa3fig6:** Determination of rs13900C/T allelic enrichment in polysome fractions and its effect on polysome loading. Polysome profile obtained by sucrose gradient centrifugation of macrophages before and after stimulation with LPS (1 µg/mL) for 3 h. (A&B) The *CCL2* mRNA shifts from monosome-associated fractions to heavier polysomes following LPS stimulation, indicating increased translation efficiency. (C&D) In contrast, the distribution of 18S shows no significant shift due to LPS treatment. (mean ± SEM, n=4). The percentage of mRNA loading on polysome was calculated using ΔCT method (mean ± SEM, n=4). (E&F) *CCL2* AEI measurement in polysomes of macrophages from heterozygous donors (n=2). Genomic and cDNA were subjected to Sanger sequencing and the peak height of both the alleles were used to determine the relative abundance of each allele.

Comment 10: Please explain in detail how primary monocytes were transfected with siRNAs for more than 72 hours. Typically, primary monocytes are very hard to transfect, have a very limited lifespan in culture (around 48 hours), and show a high level of cell death upon transfection. If monocytes were differentiated from macrophages, explain in detail how it was done and provide supporting citations from the literature.

We agree with the challenges associated with transfecting primary monocytes, including their limited lifespan in culture and susceptibility to cell death following transfection and apologize for not elaborating the method section on lentiviral transduction of primary macrophages. To overcome these limitations, we utilized monocytes undergoing differentiation into macrophages rather than fully differentiated macrophages for our experiments. Cells were transfected by slightly modifying the method described by Plaisance-Bonstaff et.al 2019 (Plaisance-Bonstaff et al., 2019). Briefly, monocytes were purified from PBMCs obtained from homozygous donors for rs13900 C or rs13900T by negative selection. Upon purification cells were resuspended in 24 well plates at a seeding density of 0.5 x10^6^ cells per well and were further cultured in the medium supplemented with 50 ng/mL M-CSF (Figure 5—figure supplement 1 and Figure 8—figure supplement 2). After 24 h, ready to use GFP-tagged pCMV6-HuR or CMV-null lentiviral particles (Amsbio, Cambridge, M.A) were transduced into 0.5 x10^6^ cells in presence of polybrene (60 µg/mL) at a MOI of 1. The cells were processed for HuR and *CCL2* expression 72 h after transduction after stimulation with LPS for 3 h. This data is now shown in new Supplementary Figure (Figure 8—figure supplement 2).

Comment 11: The authors should prove the binding of HuR to the 3′UTR of CCL2 not only in vitro but also in cells. For this aim, a CLIP including RNA fragmentation followed by RT-PCR or sequencing would be more informative than a RIP. It would be helpful also to demonstrate the different binding to the 3′UTR variants (rs13900C vs. rs13900T).

We thank the reviewer for his valuable suggestion on validating binding of HuR to the 3′UTR in cells. It is important to highlight that several independent datasets including CLIP have already demonstrated that HuR binds to the 3′UTR of *CCL2* including the region spanning the rs13900 locus. We have summarized the relevant studies in a tabular form (Supplementary Table-2). We are unable to confirm these results in new experiments due to sample limitation. The already existing data and experimental evidence provided in this manuscript strongly suggest that HuR binds within the 3′UTR. Also, a previously published study (Fan et al, 2011) showed that only the first 125 bp of the *CCL2* 3′UTR that flanks rs13900 showed strong binding to HuR but not the CCL2 coding region or other regions of 3′UTR. This further suggests that the HuR binding to the *CCL2* is localized to the 3′UTR that flanks rs13900. Please note that the primers used for amplification of the RIP material were 3′-UTR specific.

Comment 12: To quantify nascent RNA, Figure 2C should be replaced by new experiments. To label nascent RNA, authors can perform a run on/run-off experiments only with EU, without actinomycin D. As aforementioned, ActD blocks the transcription of new RNA, therefore is not useful for studying nascent RNA.

We thank the reviewer for the suggestion and would like to emphasize that while measuring the rs13900C/T allelic ratio in nascent RNA, the experimental setup included evaluating the AEI both in presence and absence of the transcriptional inhibitor actinomycin D. The data presented in Figure 2C shows that the AEI in presence of actinomycin D is amplified in comparison to non-actinomycin D treatment. This provides definitive evidence to our hypothesis that rs13900T confers greater stability to the *CCL2* message. We apologize for the oversight of not mentioning non-ACT D treatment in the methods. Necessary changes have been made to the revised manuscript (Lines 553-63).

Comment 13: The authors should also investigate the role of TIA1 as a potential RBP and explore the possibility that TIA1 may interact more with the C allele to suppress translation.

Based on the existing studies, we highlighted the importance of RNA-binding proteins such as TIA1 and U2AF56 that may interact with *CCL2* transcript (Lines 408-09). However, exploring TIA1 binding and its functional consequences are beyond the scope of the current study. We thank the reviewer for this comment and this aspect will be pursued in future studies.

Comment 14: It would be informative if the authors included study limitations and potential clinical implications of these findings, particularly regarding therapeutic approaches targeting CCL2.

We would like to inform the reviewer that the submitted manuscript included the limitations of our study. They were discussed at appropriate places and were not included as a separate section. For instance, Line 398 emphasizes the need for in-depth studies for association of rs13900 and canonical *CCL2* transcript. The need for additional studies regarding SNP-induced structural changes in RNA and its implication for RBP accessibility was highlighted at Lines 417-419. The inconclusive results of differential loading of polysomes and the need to conduct further research on the impact of rs13900 on CCL2 translatability in primary cells (Lines 457-459). We noted at Lines 484-485 about our further studies exploring the differential binding of HuR to the other regions of *CCL2* 3′UTR.

Multiple studies have indicated that functional interference of HuR as a novel therapeutic strategy, particularly in the context of cancer, inflammation, neurodegeneration, and autoimmune disorders. These approaches include inhibitors such as MS-444, KH-3, and CMLD-2 that disrupt the interaction between HuR and ARE elements or mRNAs of target genes involved in disease pathology (Chaudhary et al., 2023; Fattahi et al., 2022; Lang et al., 2017; Liu et al., 2020; Wang et al., 2019; Wei et al., 2024), offering a potential new avenue for disease treatment. Findings from our studies provide unique insights on regulation of CCL2 expression by both rs13900 and HuR. We strongly believe that the SNP rs13900 and HuR represent a new druggable target for M/M-mediated disorders such as inflammatory diseases, cancer, and cardiovascular diseases. The potential clinical implications have been discussed in the revised manuscript (Lines 487-494)

References

Barta, N., Ordog, N., Pantazi, V., Berzsenyi, I., Borsos, B.N., Majoros, H., Pahi, Z.G., Ujfaludi, Z., Pankotai, T., 2023. Identifying Suitable Reference Gene Candidates for Quantification of DNA Damage-Induced Cellular Responses in Human U2OS Cell Culture System. Biomolecules 13.

Chaudhary, S., Appadurai, M.I., Maurya, S.K., Nallasamy, P., Marimuthu, S., Shah, A., Atri, P., Ramakanth, C.V., Lele, S.M., Seshacharyulu, P., Ponnusamy, M.P., Nasser, M.W., Ganti, A.K., Batra, S.K., Lakshmanan, I., 2023. MUC16 promotes triple-negative breast cancer lung metastasis by modulating RNA-binding protein ELAVL1/HUR. Breast Cancer Res 25, 25.

Fan, J., Ishmael, F.T., Fang, X., Myers, A., Cheadle, C., Huang, S.K., Atasoy, U., Gorospe, M., Stellato, C., 2011. Chemokine transcripts as targets of the RNA-binding protein HuR in human airway epithelium. J Immunol 186, 2482-2494.

Fattahi, F., Ellis, J.S., Sylvester, M., Bahleda, K., Hietanen, S., Correa, L., Lugogo, N.L., Atasoy, U., 2022. HuR-Targeted Inhibition Impairs Th2 Proinflammatory Responses in Asthmatic CD4(+) T Cells. J Immunol 208, 38-48.

Hubal, M.J., Devaney, J.M., Hoffman, E.P., Zambraski, E.J., Gordish-Dressman, H., Kearns, A.K., Larkin, J.S., Adham, K., Patel, R.R., Clarkson, P.M., 2010. CCL2 and CCR2 polymorphisms are associated with markers of exercise-induced skeletal muscle damage. J Appl Physiol (1985) 108, 1651-1658.

Intemann, C.D., Thye, T., Forster, B., Owusu-Dabo, E., Gyapong, J., Horstmann, R.D., Meyer, C.G., 2011. MCP1 haplotypes associated with protection from pulmonary tuberculosis. BMC Genet 12, 34.

Jao, C.Y., Salic, A., 2008. Exploring RNA transcription and turnover in vivo by using click chemistry. Proc Natl Acad Sci U S A 105, 15779-15784.

Johnson, A.D., Zhang, Y., Papp, A.C., Pinsonneault, J.K., Lim, J.E., Saffen, D., Dai, Z., Wang, D., Sadee, W., 2008. Polymorphisms affecting gene transcription and mRNA processing in pharmacogenetic candidate genes: detection through allelic expression imbalance in human target tissues. Pharmacogenet Genomics 18, 781791.

Kasztelewicz, B., Czech-Kowalska, J., Lipka, B., Milewska-Bobula, B., Borszewska-Kornacka, M.K., Romanska, J., Dzierzanowska-Fangrat, K., 2017. Cytokine gene polymorphism associations with congenital cytomegalovirus infection and sensorineural hearing loss. Eur J Clin Microbiol Infect Dis 36, 1811-1818. Lang, M., Berry, D., Passecker, K., Mesteri, I., Bhuju, S., Ebner, F., Sedlyarov, V., Evstatiev, R., Dammann, K., Loy, A., Kuzyk, O., Kovarik, P., Khare, V., Beibel, M., Roma, G., Meisner-Kober, N., Gasche, C., 2017. HuR Small-Molecule Inhibitor Elicits Differential Effects in Adenomatosis Polyposis and Colorectal Carcinogenesis. Cancer Res 77, 2424-2438.

Lebedeva, S., Jens, M., Theil, K., Schwanhausser, B., Selbach, M., Landthaler, M., Rajewsky, N., 2011. Transcriptome-wide analysis of regulatory interactions of the RNA-binding protein HuR. Mol Cell 43, 340-352.

Liu, S., Huang, Z., Tang, A., Wu, X., Aube, J., Xu, L., Xing, C., Huang, Y., 2020. Inhibition of RNA-binding protein HuR reduces glomerulosclerosis in experimental nephritis. Clin Sci (Lond) 134, 1433-1448.

Mao, F., Xiao, L., Li, X., Liang, J., Teng, H., Cai, W., Sun, Z.S., 2016. RBP-Var: a database of functional variants involved in regulation mediated by RNA-binding proteins. Nucleic Acids Res 44, D154-163.

Pabis, M., Popowicz, G.M., Stehle, R., Fernandez-Ramos, D., Asami, S., Warner, L., Garcia-Maurino, S.M., Schlundt, A., Martinez-Chantar, M.L., Diaz-Moreno, I., Sattler, M., 2019. HuR biological function involves RRM3-mediated dimerization and RNA binding by all three RRMs. Nucleic Acids Res 47, 1011-1029.

Paulsen, M.T., Veloso, A., Prasad, J., Bedi, K., Ljungman, E.A., Tsan, Y.C., Chang, C.W., Tarrier, B., Washburn, J.G., Lyons, R., Robinson, D.R., Kumar-Sinha, C., Wilson, T.E., Ljungman, M., 2013. Coordinated regulation of synthesis and stability of RNA during the acute TNF-induced proinflammatory response. Proc Natl Acad Sci U S A 110, 2240-2245.

Pham, M.H., Bonello, G.B., Castiblanco, J., Le, T., Sigala, J., He, W., Mummidi, S., 2012. The rs1024611 regulatory region polymorphism is associated with CCL2 allelic expression imbalance. PLoS One 7, e49498.

Plaisance-Bonstaff, K., Faia, C., Wyczechowska, D., Jeansonne, D., Vittori, C., Peruzzi, F., 2019. Isolation, Transfection, and Culture of Primary Human Monocytes. J Vis Exp.

Ripin, N., Boudet, J., Duszczyk, M.M., Hinniger, A., Faller, M., Krepl, M., Gadi, A., Schneider, R.J., Sponer, J., Meisner-Kober, N.C., Allain, F.H., 2019. Molecular basis for AU-rich element recognition and dimerization by the HuR C-terminal RRM. Proc Natl Acad Sci U S A 116, 2935-2944.

Russo, J., Heck, A.M., Wilusz, J., Wilusz, C.J., 2017. Metabolic labeling and recovery of nascent RNA to accurately quantify mRNA stability. Methods 120, 39-48.

Wang, J., Hjelmeland, A.B., Nabors, L.B., King, P.H., 2019. Anti-cancer effects of the HuR inhibitor, MS-444, in malignant glioma cells. Cancer Biol Ther 20, 979-988.

Wei, L., Kim, S.H., Armaly, A.M., Aube, J., Xu, L., Wu, X., 2024. RNA-binding protein HuR inhibition induces multiple programmed cell death in breast and prostate cancer. Cell Commun Signal 22, 580.

Zeng, X., Zeng, W.H., Zhou, J., Liu, X.M., Huang, G., Zhu, H., Xiao, S., Zeng, Y., Cao, D., 2022. Removal of nonspecific binding proteins is required in co-immunoprecipitation with nuclear proteins. Biotechniques 73, 289-296.

Zhang, X., Chen, X., Liu, Q., Zhang, S., Hu, W., 2017. Translation repression via modulation of the cytoplasmic poly(A)-binding protein in the inflammatory response. Elife 6.